# CD1d-mediated lipid presentation by CD11c$^+$ cells regulates intestinal homeostasis

Julia Sáez de Guinoa[1,2], Rebeca Jimeno[1,2], Mauro Gaya[3], David Kipling[4], María José Garzón[5], Deborah Dunn-Walters[6], Carles Ubeda[5,7] & Patricia Barral[1,2,*]

## Abstract

Intestinal homeostasis relies on a continuous dialogue between the commensal bacteria and the immune system. Natural killer T (NKT) cells, which recognize CD1d-restricted microbial lipids and self-lipids, contribute to the regulation of mucosal immunity, yet the mechanisms underlying their functions remain poorly understood. Here, we demonstrate that NKT cells respond to intestinal lipids and CD11c$^+$ cells (including dendritic cells (DCs) and macrophages) are essential to mediate lipid presentation within the gut ultimately controlling intestinal NKT cell homeostasis and activation. Conversely, CD1d and NKT cells participate in the control of the intestinal bacteria composition and compartmentalization, in the regulation of the IgA repertoire and in the induction of regulatory T cells within the gut. These changes in intestinal homeostasis require CD1d expression on DC/macrophage populations as mice with conditional deletion of CD1d on CD11c$^+$ cells exhibit dysbiosis and altered immune homeostasis. These results unveil the importance of CD11c$^+$ cells in controlling lipid-dependent immunity in the intestinal compartment and reveal an NKT cell–DC crosstalk as a key mechanism for the regulation of gut homeostasis.

**Keywords** CD1d; microbiota; NKT cell
**Subject Categories** Immunology
**The EMBO Journal (2018) 37: e97537**

## Introduction

The mammalian intestine contains a highly complex mixture of microorganisms that perform many metabolic functions and are critical for the establishment of tissue homeostasis. The composition of the intestinal microbiota varies significantly between individuals, and evidence suggests that alterations in the populations of commensal bacteria (dysbiosis) can result in susceptibility to multiple pathological conditions including inflammatory bowel disease, obesity and diabetes. The intestinal immune system has unique features to control the expansion and composition of intestinal microbes without triggering inflammation (Hooper *et al*, 2012). Containment of commensals within the gastrointestinal tract is collaboratively managed by an epithelial layer alongside local populations of T cells, B cells, macrophages, dendritic cells (DCs) and various innate immune cells. While immune cells control the intestinal bacteria, these bacteria play a central role in educating and modulating the host immune system (Chung *et al*, 2012). For instance, it is well established that the adaptive immune system has continuous interactions with commensal bacteria to regulate the population of T-helper 17 (Th17) cells. Th17 cells are present in the intestine at steady state, and their numbers are controlled by commensal *segmented filamentous bacteria* (SFB) (Ivanov *et al*, 2009). The mechanisms by which commensals control immune cell numbers and function are complex and poorly understood, but in recent years it has become apparent that the T-cell receptor (TCR)-dependent recognition of commensal-derived antigens by intestinal T cells is a central step in the regulation of intestinal immunity. Accordingly, intestinal Th17 cells are induced in response to SFB colonization and the MHC-II-dependent presentation of SFB antigens by intestinal DCs is crucial for Th17 induction (Goto *et al*, 2014; Yang *et al*, 2014). In line with this, antigen presentation by intestinal DCs is essential to maintain intestinal homeostasis as mice with conditional deletion of MHC-II in conventional DCs develop microbial-dependent intestinal inflammation (Loschko *et al*, 2016).

As well as providing a source of protein antigens that modulate conventional T-cell immunity, commensal bacteria represent a major source of lipids, several of which have been identified for their capacity to activate a population of lipid-reactive T cells, called natural killer T (NKT) cells (Wingender *et al*, 2012; Wieland Brown *et al*, 2013; An *et al*, 2014). NKT cells have the unique

1 The Peter Gorer Department of Immunobiology, King's College London, London, UK
2 The Francis Crick Institute, London, UK
3 Ragon Institute of MGH, MIT and Harvard, Cambridge, MA, USA
4 Division of Cancer and Genetics, School of Medicine, Cardiff University, Cardiff, UK
5 Departamento de Genómica y Salud, Centro Superior de Investigación en Salud Pública – FISABIO, Valencia, Spain
6 Faculty of Health and Medical Sciences, University of Surrey, Guildford, UK
7 Centers of Biomedical Research Network (CIBER) in Epidemiology and Public Health, Madrid, Spain
*Corresponding author. Tel: +44 2037963358; E-mail: patricia.barral@kcl.ac.uk

ability to recognize through their TCRs endogenous and exogenous lipids presented on the surface molecule CD1 (CD1d in mice) (Salio *et al*, 2014). After activation, NKT cells rapidly secrete large amounts of cytokines and regulate the downstream activation of DCs, NK cells, B cells or conventional T cells ultimately modulating immune responses to infection, autoimmunity and cancer. There is growing evidence supporting a key role for intestinal NKT cells, CD1d expression and lipid presentation in the modulation of mucosal immunity in both homeostasis and disease (Dowds *et al*, 2015). NKT cells are present in the intestine of mice and humans, and CD1d is broadly expressed by intestinal cell populations including B cells, DCs, macrophages, intestinal epithelial cells (IECs) and innate lymphoid cells. NKT cells and CD1d expression have been proposed to regulate mucosal homeostasis by controlling bacterial colonization in the gut (Nieuwenhuis *et al*, 2009; Selvanantham *et al*, 2016). Conversely, commensal-derived lipids modulate the phenotype and function of intestinal NKT cells (Olszak *et al*, 2012; Wingender *et al*, 2012; An *et al*, 2014), and dysregulation of CD1d expression and NKT cell activation have been associated with the development of colitis in humans and mouse models (Perera *et al*, 2007; Liao *et al*, 2012; Olszak *et al*, 2014). Despite the wealth of data, the mechanisms controlling intestinal NKT cell function (including the antigen-presenting cells involved in lipid presentation) and their role in the modulation of intestinal homeostasis remain unknown.

The NKT cell family comprises two groups that differ in their TCR repertoire and antigen recognition: type I NKT cells (also called invariant NKT (iNKT) cells) express a semi-invariant TCR and recognize the lipid α-galactosylceramide (αGalCer). By contrast, type II NKT cells are not reactive to αGalCer and recognize mainly self-glycolipids and self-phospholipids (Salio *et al*, 2014). iNKT cells are a heterogeneous population that can be classified into several groups based on the expression of signature transcription factors: NKT1 (T-bet$^+$), NKT2 (PLZF$^{hi}$) and NKT17 (RORγt$^+$). These subsets are analogous to the conventional CD4$^+$ Th1, Th2 and Th17 subsets, but the iNKT cell subsets have been found to differentiate in the thymus where they acquire unique gene programmes that impact their features and function (Lee *et al*, 2013, 2015; Engel *et al*, 2016). Another two NKT cell subsets (NKT10 and NKT follicular helper cells) have been identified in the tissues but not in the thymus, which suggests that they may be induced in the periphery in response to antigenic stimulation (Chang *et al*, 2012; Lynch *et al*, 2015). Commensal-derived lipids have been proposed to modulate intestinal iNKT cell phenotype and numbers; however, whether lipid presentation differentially controls the homeostasis and/or development of the various iNKT cell subsets remains unexplored.

Here, we have investigated the cellular mechanisms by which NKT cells participate in the control of intestinal homeostasis. We demonstrate that CD11c$^+$ cells mediate presentation of intestinal lipids to iNKT cells regulating intestinal iNKT cell homeostasis and activation. Consequently, this CD1d-dependent DC–NKT cell crosstalk controls bacteria and immune cell populations within the intestinal compartment. Accordingly, mice with conditional deletion of CD1d on CD11c$^+$ cells display dysbiosis and altered immune homeostasis. Collectively, our studies reveal the importance of NKT cell–DC crosstalk for the regulation of intestinal homeostasis.

# Results

## CD11c$^+$ cells present intestinal lipids to iNKT cells

To investigate whether iNKT cells are recognizing intestinal lipids in steady-state conditions, we took advantage of Nur77$^{GFP}$ mice as a reporter for TCR signalling (Moran *et al*, 2011) and analysed GFP expression in iNKT cells and CD4$^+$ T cells by flow cytometry. As previously described, we observed GFP expression in CD4$^+$ T cells from all analysed tissues, since GFP expression is maintained in the steady state by tonic MHC signals (Moran *et al*, 2011; Fig 1A). By contrast, iNKT cells from spleen and liver were predominantly GFP$^-$ in agreement with their CD1d-independent survival (Matsuda *et al*, 2002; Moran *et al*, 2011). Interestingly, iNKT cells found in gut-associated tissues (including mesenteric lymph nodes (mLN) and small intestinal (SI) and colonic lamina propria (LP)) did express GFP, indicating that iNKT cells in those tissues are receiving TCR stimulation which could possibility originate from the recognition of commensal-derived lipids. To further confirm that iNKT cells can respond to intestinal lipids, we orally gavaged mice with the CD1d-restricted lipid αGalCer and measured iNKT cell activation as upregulation of Nur77 expression by intracellular staining. In WT mice, oral αGalCer induced Nur77 upregulation on mLN iNKT cells (and to a lesser extent on SI-LP iNKT cells) as detected by flow cytometry 16 h after lipid administration (Fig 1B and C, and Appendix Fig S1A). In line with this, we detected CD69 upregulation in mLN iNKT cells after αGalCer administration (Fig 1D) and quantitative PCR (qPCR) data showed increased expression of Nur77 (*Nr4a1*) in iNKT cells sort-purified from mLNs of αGalCer-treated mice (Fig 1C). Moreover, Nur77 upregulation was also detected in iNKT cells from the mLN of Nur77$^{GFP}$ mice after oral αGalCer administration (Fig 1E). Thus, altogether these data suggest that iNKT cells can respond to intestinal lipids.

Next, we investigated whether CD11c$^+$ cells are involved in CD1d-dependent presentation of intestinal lipids to iNKT cells. We took advantage of a newly generated CD1d$^{fl/fl}$ mouse (Gaya *et al*, 2018) that we crossed with CD11c-Cre mice to generate mice deficient in CD1d on CD11c$^+$ cells (CD1d$^{fl/fl}$CD11c$^{Cre}$ strain; Appendix Fig S1B–G). To characterize DCs, we identified them as CD11c$^+$MHC-II$^+$ cells and segregated the DC/macrophage populations based on the expression of CD103 and CD11b as previously described (Loschko *et al*, 2016). All three populations (CD103$^+$CD11b$^+$, CD103$^+$CD11b$^-$ and CD103$^-$CD11b$^-$) expressed CD1d in Cre$^-$ CD1d$^{fl/fl}$CD11c$^{Cre}$ mice, and the highest level of expression corresponded to CD103$^+$CD11b$^+$ DCs (Fig 1F and G, Appendix Fig S1F and G), a migratory population implicated in the transport of luminal antigens from the LP to the mLN (Bekiaris *et al*, 2014). All CD11c$^+$ populations are present in the intestinal tissues of Cre$^+$ CD1d$^{fl/fl}$CD11c$^{Cre}$ mice, and all lacked CD1d expression but expressed MHC-II, CD40, CD80 and cytokine mRNA to similar levels to CD11c$^+$ cells from Cre$^-$ CD1d$^{fl/fl}$CD11c$^{Cre}$ mice (Appendix Fig S2A–C). Strikingly, when Cre$^+$ CD1d$^{fl/fl}$CD11c$^{Cre}$ mice were orally gavaged with αGalCer, we observed a complete absence of iNKT cell activation as detected by a lack of Nur77 upregulation in iNKT cells from both mLN and SI-LP in comparison with iNKT cells from Cre$^-$ CD1d$^{fl/fl}$CD11c$^{Cre}$ mice, indicating that CD1d expression in CD11c$^+$ cells is necessary to mediate iNKT cell responses to oral lipids (Fig 1H and Appendix Fig S2D). Intriguingly, even in steady-state

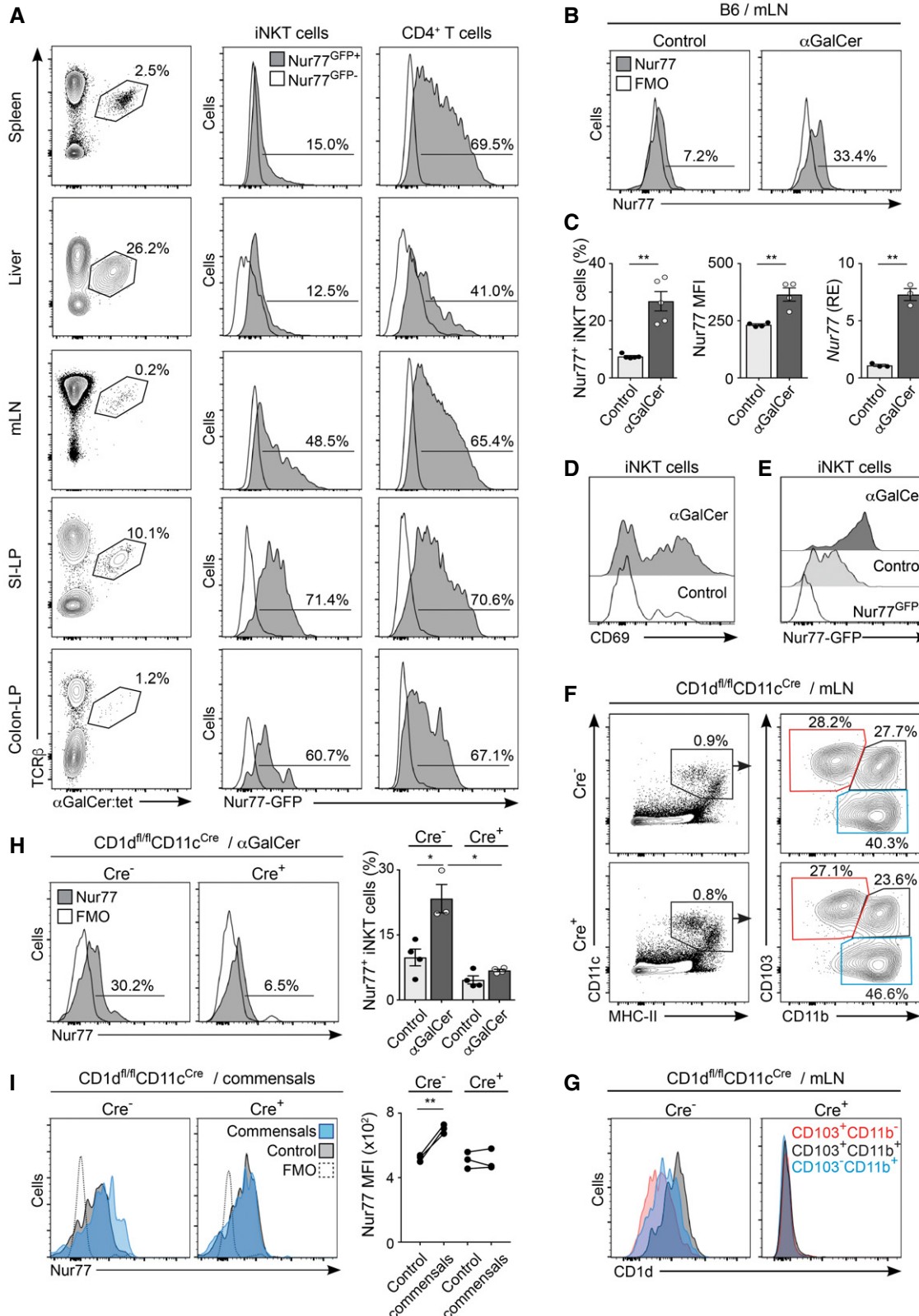

Figure 1.

**Figure 1.  Intestinal lipids presented by CD11c$^+$ cells activate iNKT cells.**

A   Flow cytometry analysis for GFP expression on iNKT (TCRβ$^+$αGalCer:tet$^+$) cells and CD4$^+$ T cells from Nur77$^{GFP+}$ reporter mice (grey histogram) and Nur77$^{GFP-}$ controls (empty histogram) in the indicated tissues. Numbers indicate percentage of iNKT cells among TCRβ$^+$ cells (left panels), of Nur77$^{GFP+}$ iNKT cells (middle) and of Nur77$^{GFP+}$CD4$^+$ T cells (right). Data represent three experiments.

B–E  WT C57BL/6 mice were orally gavaged with αGalCer or PBS, and 16 h later, mLN iNKT cells were analysed. (B) Flow cytometry profiles show intracellular Nur77 expression (grey histogram) or FMO (empty histogram) in iNKT cells from the mLN of αGalCer-treated or control mice. (C) Frequency of Nur77-expressing iNKT cells (left), Nur77 MFI (middle) and qPCR analysis of mRNA encoding Nur77 (right) in freshly isolated iNKT cells from the mLN of αGalCer-treated or control mice. qPCR data are normalized to *GAPDH*. RE, relative expression. (D) Expression of CD69 in mLN iNKT cells from control (empty histogram) or αGalCer-treated (grey histogram) mice. Data are from three experiments. (E) Flow cytometry analysis showing GFP expression on iNKT cells from Nur77$^{GFP+}$ reporter mice (grey histograms) and Nur77$^{GFP-}$ controls (empty histogram) from the mLN of αGalCer-treated or control mice.

F, G  DC populations in the mLN from Cre$^+$ and Cre$^-$ CD1d$^{fl/fl}$CD11c$^{Cre}$ mice were analysed by flow cytometry. Gating strategy (F) and CD1d expression (G) for CD103$^+$CD11b$^-$ (red), CD103$^+$CD11b$^+$ (grey) and CD103$^-$CD11b$^+$ (blue) cells. Data represent four experiments.

H   Cre$^-$ and Cre$^+$ CD1d$^{fl/fl}$CD11c$^{Cre}$ mice were orally gavaged with αGalCer, and 16 h later, Nur77 expression on iNKT cells was analysed in the mLN. Flow cytometry profiles show Nur77 (grey histogram) or FMO (empty histogram) in iNKT cells from the mLN of αGalCer-treated mice. Right panels show frequency of Nur77-expressing iNKT cells in αGalCer-treated or control Cre$^-$ and Cre$^+$ CD1d$^{fl/fl}$CD11c$^{Cre}$ mice. Data are from three experiments.

I   Single-cell suspension from mLNs from Cre$^-$ and Cre$^+$ CD1d$^{fl/fl}$CD11c$^{Cre}$ mice were prepared and incubated with commensals. Flow cytometry profiles show Nur77 expression in iNKT cells after incubation with commensals (blue histogram) or control (grey histogram). Right panels show Nur77 MFI for iNKT cells. Data are from three experiments.

Data information: Numbers indicate percentage of cells in the indicated gates. Each dot in the bar plots is an individual mouse and bars represent mean ± SEM.
*$P < 0.05$, **$P < 0.01$, two-tailed unpaired (C, H) or paired (I) *t*-test.

conditions, iNKT cells from the mLN (but not from the spleen) of Cre$^+$ CD1d$^{fl/fl}$CD11c$^{Cre}$ mice showed slightly decreased levels of intracellular Nur77 expression in comparison with iNKT cells from Cre$^-$ CD1d$^{fl/fl}$CD11c$^{Cre}$ littermates (Appendix Fig S2E). Moreover, while CD1d expression on CD11c$^+$ cells was also important to mediate activation (Nur77 upregulation) and cytokine secretion by splenic iNKT cells after i.v. injection of lipids, it was dispensable for activation of liver iNKT cells, suggesting that different APCs may mediate lipid presentation in various tissues and/or in response to different routes of antigen administration (Appendix Fig S3A). Finally, we analysed *ex vivo* whether CD1d expression on CD11c$^+$ cells is required to induce Nur77 upregulation in iNKT cells in response to commensal-derived antigens. Single-cell suspensions from the mLN of Cre$^-$ and Cre$^+$ CD1d$^{fl/fl}$CD11c$^{Cre}$ mice were prepared and incubated with commensal bacteria, and iNKT cell activation was detected as upregulation of Nur77 expression by intracellular staining. While commensal bacteria induced Nur77 upregulation in iNKT cells from Cre$^-$ CD1d$^{fl/fl}$CD11c$^{Cre}$ cultures, iNKT cell activation was absent in Cre$^+$ CD1d$^{fl/fl}$CD11c$^{Cre}$ cultures (Fig 1I). Thus, altogether our data suggest that CD1d expression in CD11c$^+$ cells is necessary to mediate iNKT cell responses to intestinal lipids.

## CD1d-dependent presentation of intestinal lipids by CD11c$^+$ cells controls the homeostasis and activation of intestinal iNKT cells

We next investigated whether lipid presentation by CD11c$^+$ cells controls intestinal iNKT cell homeostasis by analysing the iNKT cell population in the intestinal compartment of CD1d$^{fl/fl}$CD11c$^{Cre}$ mice (Fig 2A and B, Appendix Figs S3 and S4). Analyses of tissues from WT (C57BL/6) and Cre$^-$ CD1d$^{fl/fl}$CD11c$^{Cre}$ mice revealed that, as previously reported (Lee *et al*, 2013, 2015), NKT1 cells (T-bet$^+$) represent the majority of iNKT cells in thymus and spleen of mice in a C57BL/6 background (Appendix Fig S3B and C). Within the intestinal compartment, NKT1 cells were also prevalent, but we detected a significant proportion of NKT2 cells (~40%; PLZF$^{hi}$T-bet$^-$) and NKT17 cells (~5–10%; RORγt$^+$) within the mLN. Interestingly, when we analysed the iNKT cell populations in Cre$^+$ CD1d$^{fl/fl}$CD11c$^{Cre}$ mice (in comparison with Cre$^-$ CD1d$^{fl/fl}$CD11c$^{Cre}$

littermate controls), we detected a decrease in the frequency of NKT17 cells in the SI-LP, while T-bet$^+$ NKT1 cells remained unaltered (Fig 2A). In the same line, both NKT17 cells and NKT2 cells were reduced in the mLN of Cre$^+$ CD1d$^{fl/fl}$CD11c$^{Cre}$, and this was accompanied by an increase in a T-bet$^-$PLZF$^{low}$ iNKT cell population (Fig 2B). T-bet$^-$PLZF$^{low}$ iNKT cells expressed CD4 and CD49d and intermediate levels of CD44 and lacked NK1.1, PD-1, Nrp1, CD69, CD24 and the transcription factor E4BP4 (Appendix Fig S4), showing a phenotype different from that observed for NKT1, NKT2 or NKT17 cells in the mLN. These changes in iNKT cell subpopulations were already evident in the mLN of young mice which were analysed just at the time of weaning (3 weeks old) and showed a decrease in the percentage of NKT17 cells in the mLNs (Appendix Fig S5A and B). Nonetheless, iNKT cells from Cre$^+$ and Cre$^-$CD1d$^{fl/fl}$CD11c$^{Cre}$ mice retained their capacity to secrete cytokines, and after stimulation with PMA/ionomycin, up to 40% of mLN iNKT cells produced IL-4 and/or IFN-γ (Appendix Fig S5C and D). Thus, altogether these data indicate that CD1d-dependent lipid presentation by CD11c$^+$ cells can differentially control the homeostasis of the intestinal iNKT cell subsets.

To investigate whether intestinal iNKT cell subsets show differential responses to intestinal lipids, we orally gavaged mice with αGalCer and analysed the iNKT cell populations in the mLN 3 days after lipid administration (Fig 2C–H). Oral administration of lipids to WT mice resulted in an increased number of iNKT cells with a preferential expansion of the NKT2 and NKT17 populations, while the numbers of NKT1 cells remained unchanged (Fig 2C–E). In agreement with this, expression of Ki-67 (a marker for cellular proliferation) was strongly increased in NKT2 and NKT17 cells 2 days after oral lipid administration in comparison with NKT1 cells (Fig 2F), suggesting that NKT2 and NKT17 cells preferentially respond to oral lipids. Importantly, the expansion of NKT2 and NKT17 cells in response to oral lipids was dependent on CD11c$^+$ cells, as iNKT cell populations remained unaltered after oral lipid administration to Cre$^+$ CD1d$^{fl/fl}$CD11c$^{Cre}$ mice (Fig 2G and H). Thus, altogether our data indicate that CD1d-dependent presentation of intestinal lipids by CD11c$^+$ cells controls the homeostasis and activation of the intestinal iNKT cell subsets.

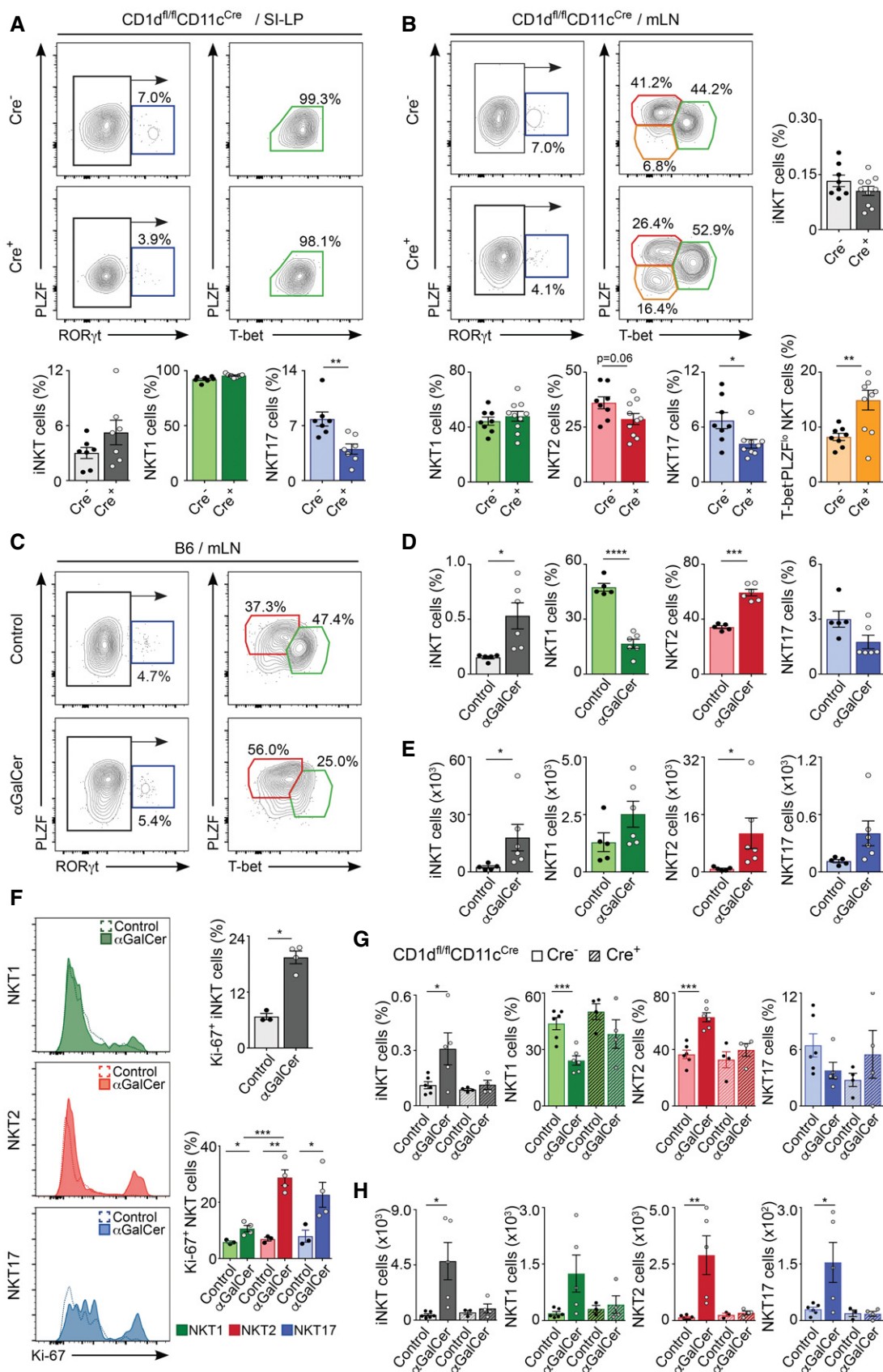

**Figure 2.**

**Figure 2. Lipid presentation by CD11c$^+$ cells controls iNKT cell subsets.**

A, B  Analysis of iNKT cells populations in the SI-LP (A) and mLN (B) from CD1d$^{fl/fl}$CD11c$^{Cre}$ mice, showing flow cytometry plots and frequency of total iNKT cells (grey) and NKT1 (RORγt$^-$PLZF$^{lo}$T-bet$^+$; green), NKT2 (RORγt$^-$PLZF$^{hi}$T-bet$^-$; red), NKT17 (PLZF$^{int}$RORγt$^+$; blue) and PLZF$^{lo}$T-bet$^-$ NKT cells (orange; mean ± SEM). Data are from 4–6 experiments.

C–E  C57BL/6 WT mice were orally gavaged with αGalCer, and iNKT cell populations were analysed 3 days later. Flow cytometry plots (C), frequency (D) and total number (E) of iNKT cells and NKT1, NKT2 and NKT17 cells in the mLN are shown. Data are from three experiments.

F  Histograms showing Ki-67 expression (left) and frequency of Ki-67$^+$ (right) NKT1, NKT2 and NKT17 cells in the mLN from C57BL/6 WT mice 2 days after oral αGalCer administration.

G, H  Frequency (G) and total number (H) of iNKT cells and NKT1, NKT2 and NKT17 cells in the mLN from Cre$^+$ and Cre$^-$ CD1d$^{fl/fl}$CD11c$^{Cre}$ mice 3 days after oral αGalCer administration. Data are from 2–3 experiments.

Data information: Numbers indicate percentage of cells in the indicated gates. Each dot in the bar plots is an individual mouse and bars represent mean ± SEM.
*$P < 0.05$, **$P < 0.01$, ***$P < 0.001$, ****$P < 0.0001$, two-tailed unpaired $t$-test.

## CD1d expression on CD11c$^+$ cells regulates the intestinal microbiota composition

Next, we investigated the functional consequences of CD1d-dependent lipid presentation for intestinal homeostasis by studying the effect of NKT cells/CD1d in the regulation of the intestinal microbiota. First, we examined the dependence of the intestinal microbial populations on CD1d/NKT cells by analysing commensal bacteria composition in CD1d-deficient mice (CD1d$^{-/-}$) obtained by crossing CD1d$^{fl/fl}$ mice with PGK-Cre mice (Appendix Fig S6). The CD1d-deficient mice resulting from these crosses showed lack of CD1d and NKT cells in all the analysed tissues, while CD1d expression levels and iNKT cell numbers were comparable in CD1d$^{+/+}$ and CD1d$^{+/-}$ mice (Appendix Fig S6). CD1d$^{-/-}$ mice were crossed with heterozygous CD1d$^{+/-}$ mice, and the CD1d$^{-/-}$ and CD1d$^{+/-}$ littermates resulting from the crosses were separated at the time of weaning (21 days after birth) into individual autoclaved cages (Ubeda *et al*, 2012). Mice were euthanized 3–4 weeks after weaning, and the composition of the microbiota colonizing the ileal epithelium, ileal lumen and caecum was determined by sequencing of bacterial 16S rDNA (Fig 3 and Appendix Fig S7A–F). We first characterized the effects of NKT cells/CD1d deficiency in the microbiota composition by using the Yue–Clayton measure of dissimilarity between bacterial communities and principal coordinates analysis (PCoA) to cluster communities along orthogonal axes of maximal variance. The first two principal coordinates of analysis (PC1 and PC2) for the ileum content separated mice based on their different genotype (CD1d$^{-/-}$ vs. CD1d$^{+/-}$), suggesting that CD1d/NKT cells participate in the regulation of the microbiota composition (Fig 3A). To determine whether specific bacterial taxa are affected by the lack of CD1d/NKT cells, we determined the relative abundance of different bacterial taxa in intestinal samples. Figure 3B shows the relative proportions of the most abundant taxa identified in CD1d$^{-/-}$ mice and CD1d$^{+/-}$ littermates, by representing the most prevalent operational taxonomic units (OTUs) comprising more than 1% of the intestinal bacteria. In our strains, most commensal bacteria belong to the *Bacteroidetes* and *Firmicutes* phyla. To identify bacterial taxa that are significantly affected by NKT cells, we used the Wilcoxon test to compare the relative abundance of specific taxa colonizing CD1d-KO and littermate control mice. To avoid false positives as the result of multiple comparisons, the Benjamini–Hochberg false discovery rate (FDR) was applied to those taxa that differed significantly ($P < 0.05$). Significant differences were observed for several OTUs with a predominant decrease in bacteria from the phylum *Bacteroidetes* (OTU04, OTU33, OTU58 and OTU123) in the intestinal lumen of

CD1d$^{-/-}$ mice vs. littermate controls (Fig 3C). Particularly substantial was the reduction observed for OTU4 (unclassified (UC) *Bacteroidales*) which represented up to 20% of the bacteria in the intestinal lumen of WT mice and was reduced to around 2% in CD1d$^{-/-}$ mice. For bacteria colonizing the intestinal epithelial wall, PCoA did not separate WT and KO mice on the basis of their genotype, but we found significant decreases in the relative frequencies of several bacteria taxa in CD1d$^{-/-}$ mice including UC *Bacteroidetes* and the family *Sutterellaceae* (Fig 3A and D). No differences were found in SFB (which are known to colonize the ileum wall) between WT and KO mice, by deep sequencing or qPCR (Appendix Fig S7C). Added to this, no significant differences were found in the Shannon diversity index between WT and KO mice, suggesting that CD1d/NKT cells do not influence the total diversity of the intestinal microbiota (Appendix Fig S7B). In the caecum, we measured a decrease in the total number of bacteria in CD1d$^{-/-}$ mice, but we did not detect any significant differences in any bacterial taxa between CD1d$^{-/-}$ and CD1d$^{+/-}$ mice (Appendix Fig S7D and E).

To further confirm the role of CD1d/NKT cells in the regulation of the intestinal microbiota, we orally gavaged WT mice with αGalCer and analysed the microbial populations in the stool of the mice before (day 0, d0) and 10 days (d10) after lipid administration. As shown in Fig 3E–G, oral lipids induced strong changes in the microbial communities and PCoAs separated samples based on lipid administration (d0 vs. d10). Microbial changes can be detected at the level of phylum, as αGalCer administration resulted in an increase in *Bacteroidetes* and *Proteobacteria* and a decrease in *Firmicutes* and *Deferribacteres* (Fig 3G). Accordingly, at the OTU level, we detected a significant decrease in bacteria belonging to the *Firmicutes* phylum (i.e. *Oscillibacter, Flavonifractor,* UC Lachnospiraceae) and an increase in OTUs belonging to the *Bacteroidetes* phylum (i.e. UC Bacteroidales; Appendix Fig S7F). It is worth noting that while CD1d-dependent iNKT cell activation resulted in an increase in *Bacteroidetes* (Fig 3F and G), mice deficient in CD1d/NKT cells showed the opposite phenotype with a decrease in OTUs belonging to the phylum *Bacteroidetes* (Fig 3B and C). Thus, altogether our data demonstrate that NKT cells and CD1d expression contribute to the regulation of the intestinal bacteria composition.

Having shown that CD1d/NKT cells shape the intestinal microbiota, we next examined whether CD1d-dependent DC–NKT cell crosstalk is required to control the populations of intestinal microbes. Pups from the CD1d$^{fl/fl}$CD11c$^{Cre}$ strain were separated at the time of weaning in individual cages, and the composition of the microbiota colonizing the ileal lumen was analysed 3 weeks later. Analyses of the ileum content in the CD1d$^{fl/fl}$CD11c$^{Cre}$ strain showed

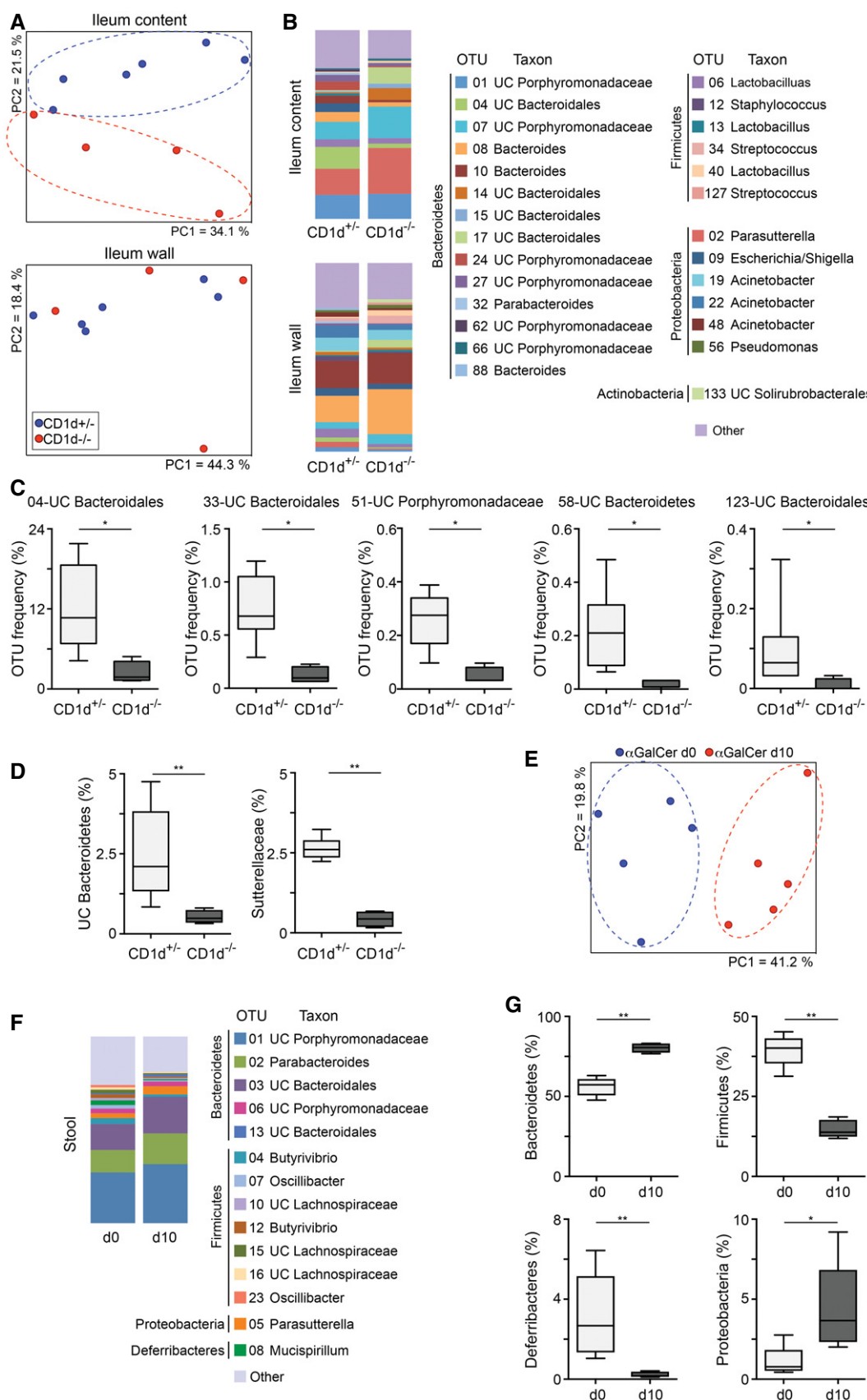

Figure 3.

**Figure 3. CD1d and NKT cells regulate the intestinal microbiota.**

A   Principal coordinates analysis (PCoA) using the Yue–Clayton distances obtained for bacterial samples from the ileum content and ileum wall of CD1d$^{+/-}$ and CD1d$^{-/-}$ mice. The axes show the percentage of variation explained by PC1 and PC2. Each dot corresponds to one mouse.

B   Average relative abundance of the most frequent (> 1%) operational taxonomic units (OTUs) of the ileum content and ileum wall from CD1d$^{+/-}$ and CD1d$^{-/-}$ mice. Bacterial taxa (at the genus level, or the closest level of classification) are shown, grouped by phylum and labelled with different colours as indicated. UC, unclassified.

C, D  Relative abundance of specified OTUs in the ileum content (C) and of specified taxa in the ileum wall (D) from CD1d$^{+/-}$ and CD1d$^{-/-}$ mice.

E–G  C57BL/6 mice were orally administered αGalCer, and faecal bacteria were analysed before (d0) and 10 days (d10) after the treatment. (E) PCoA using the Yue–Clayton distances obtained among faecal samples at d0 and d10. The axes show the percentage of variation explained by PC1 and PC2. Each dot corresponds to one mouse. (F) Average relative abundance of the most frequent (> 1%) OTUs at d0 and d10. Taxa are shown and labelled with different colours as indicated. (G) Relative abundance of the specified phyla, before and 10 days after αGalCer treatment.

Data information: In the boxplots, lines indicate the median, boxes show the 75$^{th}$ and the 25$^{th}$ percentiles and whiskers indicate the maximum and minimum values. *$P < 0.05$, **$P < 0.01$, two-tailed Wilcoxon test.

differences in bacterial taxa between Cre$^-$ and Cre$^+$ mice (Fig 4A and B). Accordingly, Cre$^+$ CD1d$^{fl/fl}$CD11c$^{Cre}$ mice showed significant differences in several OTUs with a predominant decrease in OTUs belonging to the phylum Bacteroidetes (OTU04, OTU03 and OTU17) in the intestinal lumen vs. Cre$^-$ CD1d$^{fl/fl}$CD11c$^{Cre}$ littermate controls (Fig 4A and B). It is worth noting that the microbiota composition was different between the CD1d$^{-/-}$ and CD1d$^{fl/fl}$CD11c$^{Cre}$ strains probably due to the dominance of maternal transmission in the shaping of the intestinal microbiota (Ubeda *et al*, 2012). However, some changes in bacterial taxa were conserved. For instance, OTU04 (UC *Bacteroidales*) was highly prevalent in both CD1d$^{+/-}$ and Cre$^-$ CD1d$^{fl/fl}$CD11c$^{Cre}$ mice and strikingly reduced in the CD1d-deficient mice from both strains (Figs 3C and 4B). Thus, CD1d expression on CD11c$^+$ cells contributes to the shaping of the intestinal bacterial population.

## CD1d/NKT cells control intestinal bacteria compartmentalization

One of the key mechanisms to maintain intestinal homeostasis is to minimize the contact between bacteria and the epithelial surface. Since the composition of intestinal bacteria is altered in CD1d-deficient mice, we analysed the compartmentalization of intestinal bacteria in this strain. To do this, we performed RNAscope, an *in situ* hybridization technique that allows specific labelling of bacteria and visualization together with intestinal cells detected by haematoxylin staining. As shown in Fig 4C, while in CD1d$^{+/-}$ mice commensal bacteria were physically separated from the IECs, intestinal microbes are not efficiently segregated in the small intestine of CD1d$^{-/-}$ mice and bacteria are found in direct contact with the intestinal epithelium. Added to this, mLN of CD1d$^{-/-}$ mice (but not spleen) showed increased cellularity in comparison with littermate controls, which may suggest increased penetration of bacteria and/or bacterial products from the intestinal lumen (Fig 4D). In the same line, oral infection of mice with *Yersinia enterocolitica* (which adhere to epithelial and M cells and translocate through IECs) resulted in an increased number of bacteria found in the mLN of CD1d$^{-/-}$ mice in comparison with littermate controls (Fig 4E). Despite these phenotypes, CD1d$^{-/-}$ mice showed normal gross intestinal morphology, goblet cell number and mucin and antimicrobial peptide mRNA levels in IECs (Appendix Fig S8). In contrast to CD1d$^{-/-}$ mice, RNAscope analyses of the intestinal ileum of CD1d$^{fl/fl}$CD11c$^{Cre}$ mice showed commensal bacteria physically separated from the intestinal epithelium both in Cre$^+$ CD1d$^{fl/fl}$CD11c$^{Cre}$ mice and in Cre$^-$ CD1d$^{fl/fl}$CD11c$^{Cre}$ littermate controls

(Fig 4F). Thus, altogether our data demonstrate that NKT cells and CD1d expression contribute to maintain the spatial segregation between microbes and IECs through a mechanism independent of lipid presentation by CD11c$^+$ cells.

## Altered IgA repertoire in CD1d-deficient and CD1d$^{fl/fl}$CD11c$^{Cre}$ mice

One of the immune mechanisms central to the maintenance of intestinal homeostasis is the production of IgA by intestinal B cells. We and others have established the capacity of NKT cells to modulate B-cell responses (Barral *et al*, 2008; Leadbetter *et al*, 2008; Chang *et al*, 2012; King *et al*, 2012), yet their potential to influence IgA production within the intestine remains unexplored. Consequently, we investigated whether NKT cells/CD1d modulate intestinal B-cell responses and IgA production. To evaluate the impact of NKT cell/CD1d deficiency in intestinal IgA production, we analysed the quantity and quality of IgA in CD1d$^{-/-}$ mice and littermate controls. While the concentration of free IgA in the faeces and blood was comparable between CD1d-deficient and control mice, the repertoire of the IgA differed (Fig 5A and B). When we sequenced the immunoglobulin heavy chain (IgH) of IgA-producing cells, we observed that both CD1d$^{+/-}$ and CD1d$^{-/-}$ mice have a polyclonal repertoire, yet CD1d-deficient mice had an enrichment of IgA-producing cells with the variable region of the IgH locus belonging to the *IGHV1* family. Most of the analysed sequences showed evidence of antigen-driven mutation; however, we did not observe major differences in the ratio of mutations in any of the complementarity-determining regions (CDRs) or framework regions (FWRs), or in the calculated affinity maturation index (Appendix Fig S9). In the same line, analyses of the IgA repertoire in the CD1d$^{fl/fl}$CD11c$^{Cre}$ strain showed significant differences between Cre$^-$ and Cre$^+$ mice with an increase in the *IGHV5* family usage in Cre$^+$ CD1d$^{fl/fl}$CD11c$^{Cre}$ mice (Fig 5C). Thus, these data indicate that CD1d expression in CD11c$^+$ cells is important for the control of the intestinal IgA repertoire.

The altered repertoire of IgA$^+$ cells in CD1d-deficient mice and CD1d$^{fl/fl}$CD11c$^{Cre}$ mice could be because of intrinsic factors (a defect in B-cell development and/or activation) or, conversely, because changes in the intestinal microbiota could ultimately result in alterations in the IgA production and/or repertoire. Analyses of B-cell populations in spleen, bone marrow and peritoneal cavity showed no differences in frequencies or absolute numbers of B-cell populations between CD1d$^{+/-}$ and CD1d$^{-/-}$ mice (Appendix Fig S10A). Similarly, when we analysed the intestinal compartment, we

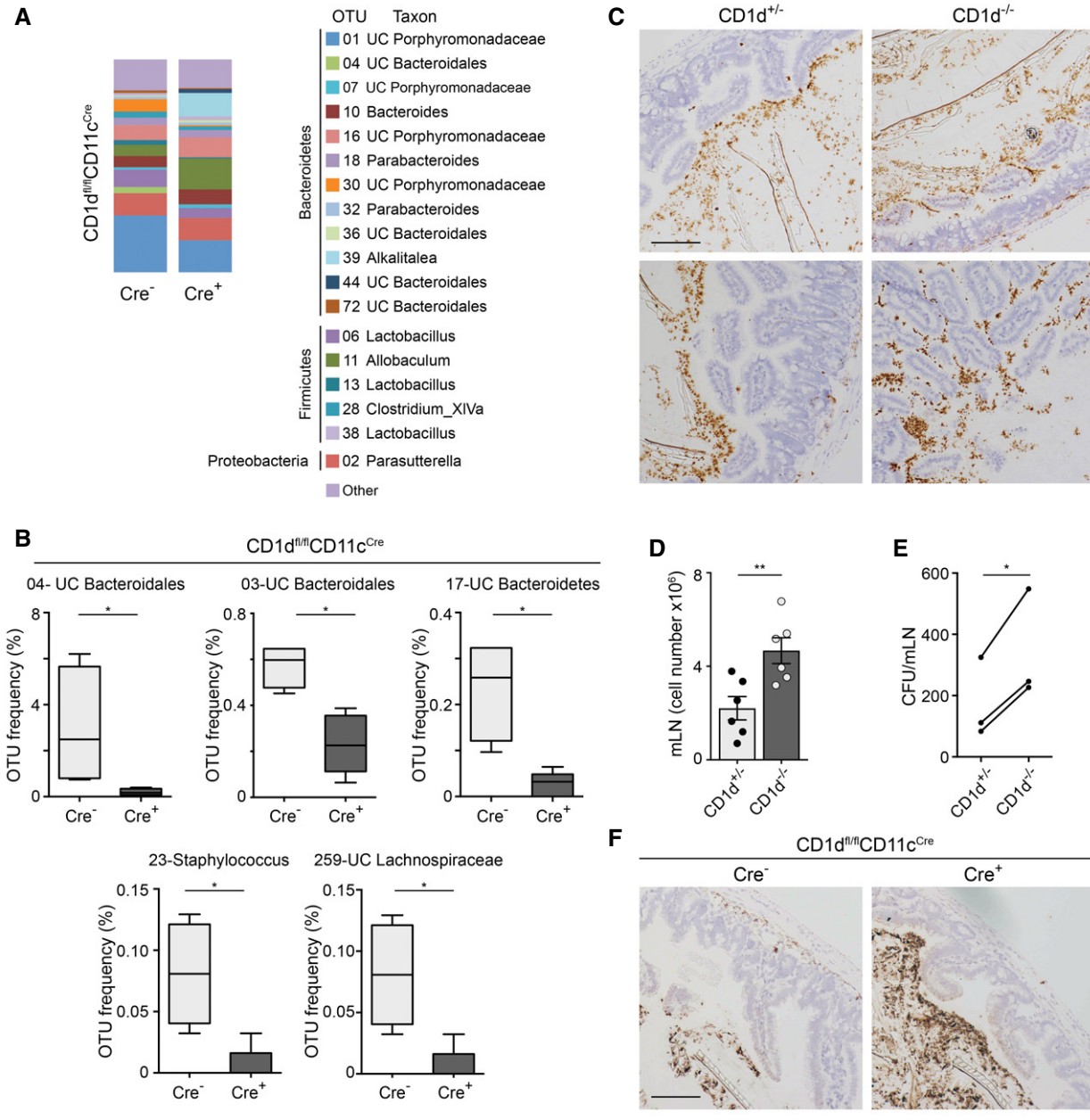

**Figure 4.  CD1d expression on CD11c⁺ cells shapes the intestinal microbiota.**

A   Average relative abundance of the most frequent (> 1%) OTUs in the ileum content from Cre⁻ and Cre⁺ mice of the CD1d^fl/fl^CD11c^Cre^ strain (*n* = 4–5). Bacterial taxa (at the genus level, or the closest level of classification) are shown, grouped by phylum and labelled with different colours as indicated.

B   Relative abundance of the specified OTUs in the ileum content from Cre⁻ and Cre⁺ CD1d^fl/fl^CD11c^Cre^ mice (*n* = 4–5). Lines indicate the median, boxes show the 75^th^ and the 25^th^ percentiles and whiskers indicate the maximum and minimum values. *$P < 0.05$, two-tailed Wilcoxon test.

C   Localization of bacteria (brown) and intestinal cells (blue), detected by RNAscope and haematoxylin staining, respectively, in ileum sections from CD1d^+/−^ and CD1d^−/−^ mice. Data represent two experiments. Scale bar, 100 μm.

D   Total cell numbers in the mLN of CD1d^+/−^ and CD1d^−/−^ mice. Each dot in the bar plots is an individual mouse and bars represent mean ± SEM. **$P < 0.01$, two-tailed unpaired *t*-test.

E   *Yersinia enterocolitica* translocation to the mLNs from CD1d^+/−^ and CD1d^−/−^ mice orally infected with *Y. enterocolitica* ($1 \times 10^9$ CFU), assessed 24 h after by selective plating. Data are pooled from three experiments (*n* = 9). *$P < 0.05$, two-tailed paired *t*-test.

F   Localization of bacteria (brown) and intestinal cells (blue), detected by RNAscope and haematoxylin staining in ileum sections from Cre⁻ and Cre⁺ CD1d^fl/fl^CD11c^Cre^ mice. Data represent two experiments. Scale bar, 100 μm.

observed that the frequencies and numbers of IgA-producing cells in the LP and Peyer's patches (PP) were comparable in CD1d^−/−^ and CD1d^+/−^ mice (Appendix Fig S10A). Moreover, *in vitro* stimulation experiments showed a similar capacity for CD1d-deficient and CD1d-sufficient B cells to proliferate and class-switch in response to BCR and/or TLR stimulation (Appendix Fig S10B–D). Finally, oral

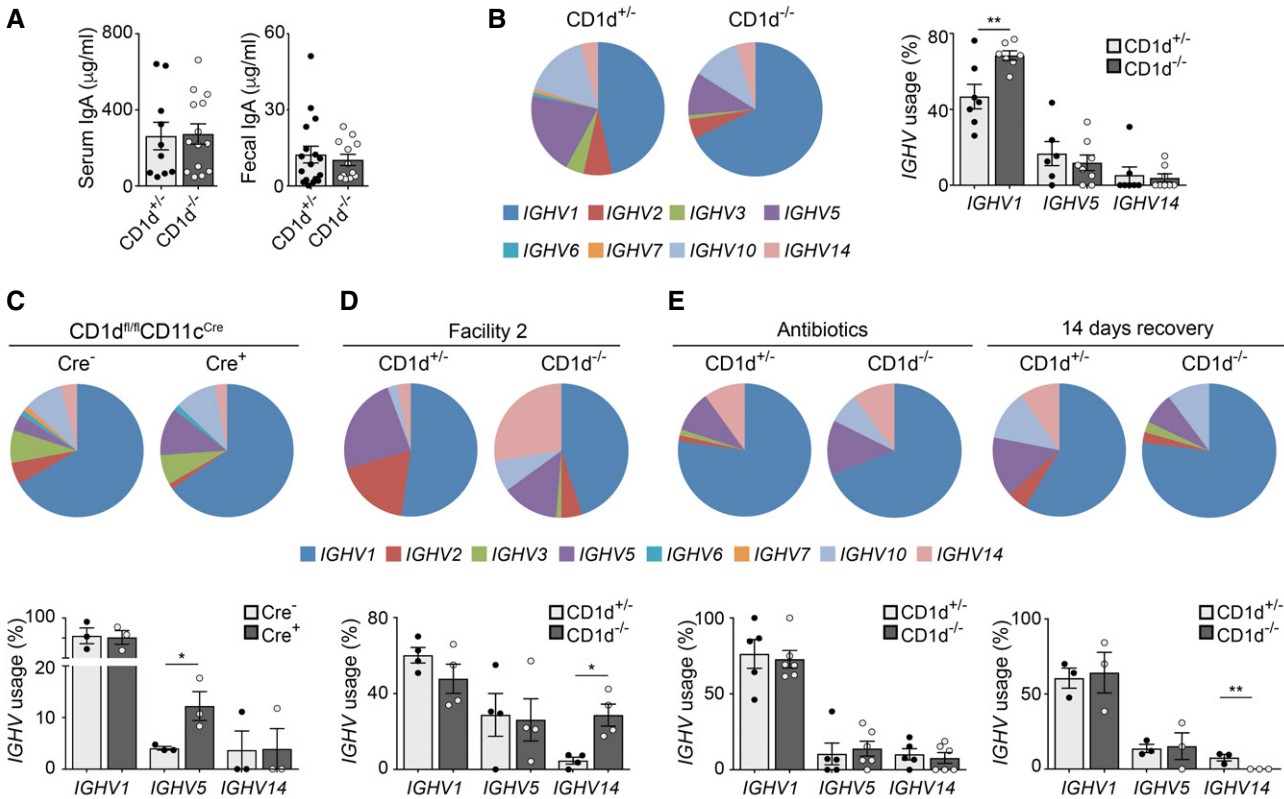

**Figure 5. CD1d expression on CD11c⁺ cells modulates the IgA repertoire.**

A   IgA levels in serum and faeces from CD1d⁺/⁻ and CD1d⁻/⁻ mice, determined by ELISA. Data are from more than five experiments.

B   IGHV family usage (left) and frequency of representative IGHV families (right) for IgA in Peyer's patches from CD1d⁺/⁻ and CD1d⁻/⁻ mice. Data are from four experiments.

C   IGHV family usage (top) and frequency of most frequent IGHV families (bottom) for IgA in Cre⁻ and Cre⁺ CD1d^fl/fl CD11c^Cre mice. Data are from three experiments.

D   IGHV family usage (top) and frequency of representative IGHV families (bottom) for IgA in CD1d⁺/⁻ and CD1d⁻/⁻ mice from a different animal facility. Data are from two experiments.

E   IGHV family usage (top) and frequency of most frequent IGHV families (bottom) for IgA in CD1d⁺/⁻ and CD1d⁻/⁻ mice treated with antibiotics for 14 days (left) and 14 days after the treatment (right). Data are from two experiments.

Data information: Each dot in the bar plots is an individual mouse and bars represent mean ± SEM. *P < 0.05, **P < 0.01, two-tailed unpaired t-test.

challenge of CD1d⁺/⁻ and CD1d⁻/⁻ mice with cholera toxin induced strong cholera toxin-specific IgA responses detectable in serum and faeces (Appendix Fig S10E). Thus, altogether these results suggest that B-cell development and activation are unaltered in CD1d-deficient mice.

It is well established that the majority of IgA responses in the intestine are driven by the microbiota (Hapfelmeier *et al*, 2010) and, since the B-cell population is normal in CD1d-deficient mice, it is conceivable that the changes in the IgA repertoire are a result of the altered microbiota observed in CD1d⁻/⁻ and Cre⁺ CD1d^fl/fl CD11c^Cre mice. In agreement with this hypothesis, we detected a different IgA repertoire in CD1d⁺/⁻ from a different animal facility while still preserving the differences between CD1d⁺/⁻ and CD1d⁻/⁻ mice (Fig 5D). To formally test whether changes in IgA are a result of dysbiosis, we reduced intestinal commensal populations by orally gavaging CD1d⁻/⁻ and control mice with a mixture of antibiotics for 14 days and analysed IgA repertoire after treatment. Antibiotic administration led to a 1,000-fold decrease in the 16S rDNA copy number as quantified in stool samples using qPCR (Appendix Fig

S10F). As a result of the elimination of commensal bacteria, the differences in IgA repertoire between CD1d⁺/⁻ and CD1d⁻/⁻ mice disappeared, and no significant differences were observed in IGHV family usage (Fig 5E). However, when mice were treated with antibiotics and then left untreated for 2 weeks to recover their microbiota, we again observed significant differences in the IGHV usage in IgA from CD1d⁺/⁻ and CD1d⁻/⁻ mice (Fig 5E). Thus, altogether these experiments suggest that CD1d deficiency results in changes in the microbiota that in turn give rise to an altered IgA repertoire.

## NKT cells regulate intestinal immunity by controlling Treg induction

Natural killer T cells have been shown to regulate the function of other immune cells in steady-state conditions, including memory CD8⁺ T cells in the thymus (Lee *et al*, 2013) and Treg numbers and function in adipose tissue (Lynch *et al*, 2015). Consequently, we investigated whether intestinal NKT cells participate in the

regulation of the homeostasis and/or function of intestinal T cells. To determine whether NKT cells control steady-state T-cell populations, we analysed the populations of Tregs, $CD4^+$ and $CD8^+$ T cells, in the intestinal tissues of CD1d-deficient mice. While the frequencies of naïve and activated $CD4^+$ and $CD8^+$ T cells remained unaltered (Appendix Fig S11), their numbers increased due to the total increased cellularity of mLN in $CD1d^{-/-}$ mice (Fig 3D and Appendix Fig S11A). However, we detected lower frequencies of peripherally induced Tregs (iTregs as defined by Nrp1) in the mLN and SI-LP of $CD1d^{-/-}$ mice vs. littermate controls (Fig 6A and Appendix Fig S11B), suggesting that NKT cells could contribute to the regulation of the intestinal Treg population in steady-state conditions. In agreement with this hypothesis, activation of iNKT cells by oral administration of αGalCer to WT mice resulted in an increased frequency and numbers of iTregs in mLN (but not spleen) at 3 days after lipid administration (Fig 6B and C, Appendix Fig S11C), while the Treg population remained unchanged after administration of oral lipids to $Cre^+$ $CD1d^{fl/fl}CD11c^{Cre}$ mice (Fig 6D and E). To provide mechanistic insight into the processes by which iNKT cell activation modulates the Treg population, we analysed the pattern of cytokine secretion in the mLN after oral lipid administration. We have previously demonstrated that NKT2 cells [which are known to produce high levels of IL-4 (Lee *et al*, 2013)] expand in the mLN of αGalCer-treated mice but not in $Cre^+$ $CD1d^{fl/fl}CD11c^{Cre}$ mice (Fig 2C–H). Accordingly, qPCR analyses of sorted iNKT cells revealed that mLN iNKT cells produce IL-4 after αGalCer administration (Fig 6F). Added to this, we detected increased levels of TGF-β mRNA in the mLN of αGalCer-treated WT mice but not in $CD1d^{-/-}$ or $Cre^+$ $CD1d^{fl/fl}CD11c^{Cre}$ mice (Fig 6G). Since both TGF-β and IL-4 have been suggested to regulate the proliferation and/or survival of Tregs (Chen *et al*, 2003; Skapenko *et al*, 2005; Pillai *et al*, 2009), we analysed whether these cytokines could control Treg induction after iNKT cell activation *in vivo*. While administration of an αTGF-β blocking antibody *in vivo* did not affect iTregs, blocking IL-4 prevented iTreg induction after αGalCer administration (Fig 6H and I), yet none of the blocking antibodies affected iNKT cell activation (Appendix Fig S11D). Importantly, iNKT cells are major producers of IL-4 in the mLN (Fig 6F, J, and K). Accordingly, after stimulation of mLN lymphocytes with PMA/ionomycin, iNKT cells constitute up to 60% of the IL-4-producing cells (Fig 6J). Moreover, while mLN iNKT cells produced IL-4 after αGalCer administration (Fig 6F), we did not detect changes in *IL4* mRNA levels in $CD11c^+$ cells, B cells or T cells from the mLN of αGalCer-treated mice (Fig 6K), although we cannot discard that other mLN cell populations may also contribute to IL-4 secretion in this context. Thus, altogether our data indicate that NKT cells contribute to the regulation of the intestinal Treg population through production of IL-4.

## Discussion

Intestinal NKT cells play a central role in the regulation of mucosal immunity, and dysregulation of CD1d expression and NKT cell activation have been associated with the development of intestinal inflammation in mice and humans (Dowds *et al*, 2015). Thus, the strict control of CD1d expression and NKT cell responses is a major component in the establishment of intestinal homeostasis and in the regulation of intestinal immunity. The pathways and mechanisms that regulate intestinal NKT cell functions are therefore of considerable importance but have remained poorly understood mainly due to our limited knowledge of the properties and functions of NKT cells within specific tissues, and how those relate to the expression patterns of CD1d. Here, we found that CD1d-dependent presentation of intestinal lipids by $CD1c^+$ cells is essential to control the intestinal iNKT cell population ultimately contributing to the regulation of intestinal homeostasis (Appendix Fig S12). Accordingly, our data reveal that both CD1d-KO mice and mice lacking CD1d on $CD11c^+$ cells show dysregulated intestinal microbiota composition and skewed IgA repertoire. However, while CD1d-KO mice showed altered commensal bacteria compartmentalization, this was not evident in $CD1d^{fl/fl}CD11c^{Cre}$ mice, suggesting that additional mechanisms (and probably other $CD1d^+$ APCs and/or CD1d-independent processes) may regulate the architecture of commensal bacteria within the intestinal compartment.

The iNKT cell populations have been proposed to differentiate in the thymus and to possibly recognize different lipids which may determine their development and function (Lee *et al*, 2013). However, it is also possible that the presence of the iNKT cell subpopulations in the periphery is controlled by tissue-specific clues, which could dictate the homing and/or proliferation of specific iNKT cell families in various tissues. In agreement with this hypothesis, our data show that lipid presentation by $CD11c^+$ cells differentially controls the homeostasis and activation of the various iNKT cell families within the intestinal compartment. Importantly, iNKT cells have been shown to proliferate in the intestine of newborn mice in a process regulated by exposure to commensals (Olszak *et al*, 2012; An *et al*, 2014). Consequently, it is possible that the local presentation of commensal-derived lipids by $CD11c^+$ cells differentially controls the proliferation and/or differentiation of the iNKT cell families explaining the altered NKT cell populations in $CD1d^{fl/fl}CD11c^{Cre}$ mice. Although the mechanisms by which the various iNKT cell subsets respond differently to oral lipids will require further investigation, it has been suggested that the iNKT families occupy different anatomical locations within the mLN (Lee *et al*, 2015), which could result in different capacity to interact with lipid-bearing DCs arriving to the LN.

Although we did not detect major changes in the numbers of intestinal iNKT cells in $CD1d^{fl/fl}CD11c^{Cre}$ mice, we did observe alterations in the intestinal iNKT cell populations including a decrease in NKT17 cells in both the mLN and SI-LP. Interestingly, a similar phenotype has been described in germ-free mice that show a reduction in $CD127^+CD4^-$ iNKT cells in the SI-LP (Wingender *et al*, 2012) (of note, $CD127^+CD4^-$ iNKT cells correspond to $ROR\gamma t^+$ NKT17 cells in our experiments). Moreover, $CD127^+CD4^-$ iNKT cell numbers are different in WT mice from different vendors (Wingender *et al*, 2012), indicating that exposure to different commensals can ultimately control the populations of intestinal iNKT cells. Intriguingly, although SFB have been suggested to be the main drivers for intestinal Th17 differentiation (Ivanov *et al*, 2009), our microbiota sequencing experiments did not detect significant differences in SFB in any of our strains, which suggests that the intestinal NKT17 cell population may be controlled by different bacterial species to those inducing conventional Th17 differentiation.

Intestinal homeostasis relies on a continuous dialogue between the commensal bacteria and the intestinal immune system. Many factors contribute to the control of the intestinal microbiota

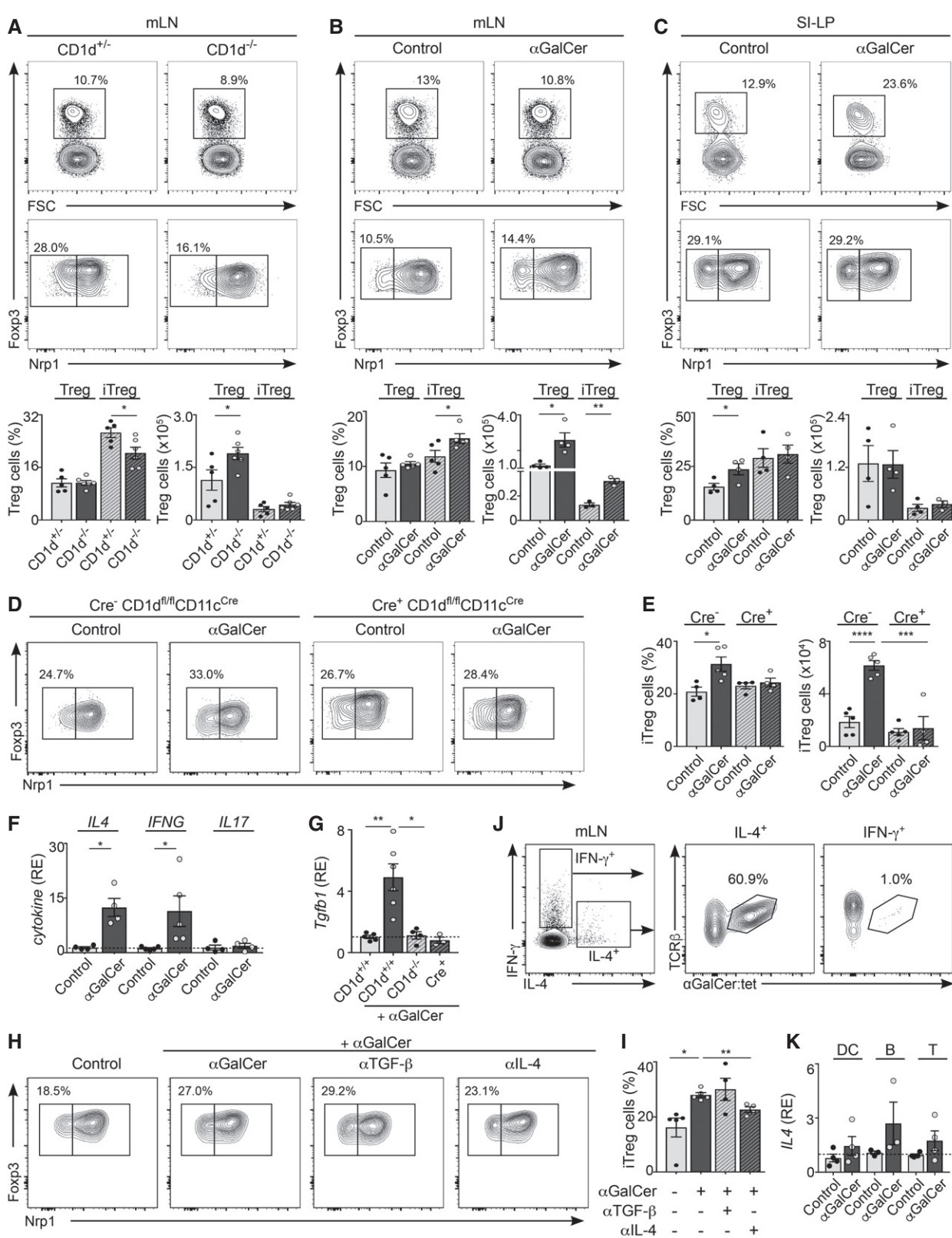

**Figure 6.**

composition, including familial transmission and diet, but also genetics. For instance, MHC polymorphisms can significantly alter intestinal microbial communities, suggesting that specific T-cell recognition of commensal-derived antigens strongly contributes to the establishment of intestinal homeostasis (Kubinak *et al*, 2015). Previous studies proposed a role for CD1d/NKT cells in the control

◄

**Figure 6.  iNKTcells/CD1d regulate intestinal immunity by controlling Treg induction.**

A       Regulatory T cells (Tregs; CD4$^+$Foxp3$^+$) and induced Tregs (iTregs; Nrp1$^-$) in the mLN from CD1d$^{-/-}$ and CD1d$^{+/-}$ mice. Flow cytometry plots (top) and frequency and numbers of Treg and iTreg (bottom) are depicted. Data are from three experiments.

B, C    C57BL/6 WT mice were orally gavaged with αGalCer, and the Treg populations were analysed 3 days later in mLN (B) and SI-LP (C). Flow cytometry plots (top) and frequency and numbers of Treg and iTreg (bottom) are depicted. Data are from three independent experiments.

D, E    Cre$^+$ and Cre$^-$ CD1d$^{fl/fl}$CD11c$^{Cre}$ mice were orally gavaged with αGalCer, and the Treg populations were analysed 3 days later in mLN. Flow cytometry plots (D) and frequency and numbers of iTregs (E) are depicted. Data are from 2–3 experiments.

F       qPCR analyses for the depicted cytokines in freshly isolated iNKT cells from the mLN of C57BL/6 αGalCer-treated or control mice. Data are normalized to *GAPDH*. RE, relative expression. Data are from 3–4 experiments.

G       qPCR analyses for *Tgfb1* in the mLN of CD1d$^{+/+}$, CD1d$^{-/-}$ or Cre$^+$ CD1d$^{fl/fl}$CD11c$^{Cre}$ mice after oral αGalCer administration. Data are from three experiments.

H, I    C57BL/6 WT mice were orally gavaged with αGalCer ± αIL-4 or αTGF-β blocking antibodies, and the Treg population was analysed 3 days later in mLN. Flow cytometry plots (H) and frequency of iTregs (I) are depicted. Data are from three experiments.

J       Single-cell suspensions from the mLN of WT mice were stimulated with PMA/ionomycin for 3 h, and secretion of IFN-γ and IL-4 was measured by flow cytometry. Plots show production of IFN-γ and IL-4 by mLN cells (left) and αGalCer-loaded CD1d-tetramer binding for IL-4$^+$ (middle) or IFN-γ$^+$ (right) cells. Data are representative from three experiments.

K       qPCR analyses for *IL4* mRNA in freshly isolated CD11c$^+$ cells, B cells and T cells sort-purified from the mLN of WT αGalCer-treated or control mice. Data are normalized to *HPRT-1*. RE, relative expression. Data are from four experiments.

Data information: Numbers indicate percentage of cells in the indicated gates. Each dot in the bar plots is an individual mouse and bars represent mean ± SEM.
*$P < 0.05$, **$P < 0.01$, ***$P < 0.001$, ****$P < 0.0001$, two-tailed unpaired *t*-test.

of the intestinal bacteria composition as microbial communities were found to be different in WT and CD1d-KO mice (Nieuwenhuis *et al*, 2009). Our studies confirm and extend these observations as we provide evidence that CD1d-mediated presentation of intestinal lipids shapes the intestinal microbiota and that CD1d deficiency in CD11c$^+$ cells is sufficient to modify the commensal communities colonizing the ileum. A variety of immune mechanisms have been proposed to control the intestinal microbiota including IgA, anti-microbial peptides (AMP) and mucus (Hooper *et al*, 2012). Although our experiments did not find differences in AMP mRNA in CD1d$^{+/+}$ and CD1d$^{-/-}$ IECs, it has been suggested that iNKT cells could modulate Paneth cell degranulation (and consequently AMP secretion) through production of IFN-γ (Farin *et al*, 2014). Alternatively, several additional mechanisms have been reported to contribute to the regulation of intestinal bacterial communities. For instance, different microbiota can provide different qualities to the mucus layer, affecting the penetrability of bacteria (Jakobsson *et al*, 2015). Moreover, alterations in the mucus layer as a consequence of altered glycosylation can result in changes in the intestinal microbial populations (Sommer *et al*, 2014). Although further research is needed, it seems likely that DC-mediated activation of intestinal NKT cells could directly (through cytokine secretion) or indirectly (through modulation of other immune cell populations, i.e. Tregs) influence IEC function ultimately leading to alterations in the intestinal microbiota composition and compartmentalization.

Although we found that all of the intestinal DC/macrophage populations express CD1d, the level of expression is different in the various populations, being higher in CD103$^+$CD11b$^+$ DCs in the mLN, but not in the SI-LP. Interestingly, CD103$^+$CD11b$^+$ DCs have been described as a main migratory population that arrives to the mLN from the LP (Bekiaris *et al*, 2014). Consequently, the differences in CD1d expression in the various tissues could be due to the different environmental conditions as cells in the LP may be exposed to epithelial-derived cytokines and/or microbiota-derived metabolites that could modulate CD1d expression levels. Nonetheless, all of the CD11c$^+$ cells in mLN and SI-LP express CD1d and they could potentially be involved in lipid presentation to iNKT cells in the various anatomical locations and/or at different stages of the immune responses. In addition to its role in lipid

presentation, CD1d has been shown to modulate APC function. Accordingly, CD1d crosslinking is sufficient to induce activation and secretion of cytokines by various APCs. For instance, ligation of CD1d in monocytes leads to NF-κB-dependent IL-12 production (Yue *et al*, 2005), while engagement of CD1d in group 3 innate lymphoid cell results in IL-22 secretion (Saez de Guinoa *et al*, 2017). Added to this, engagement of CD1d in IECs results in STAT3-dependent IL-10 secretion providing protective effects in murine models of IBD (Olszak *et al*, 2014). Although we did not detect any major phenotypical differences between CD1d-sufficient and CD1d-deficient intestinal DCs in steady-state conditions, we cannot discard the possibility that CD1d expression itself may modulate DC functions which may in turn control other intestinal immune cells.

Natural killer T cells have been proposed to modulate the function of Tregs in various tissues contributing to the regulation of tissue homeostasis and/or controlling disease progression. For instance, in steady state, NKT cells induce an anti-inflammatory phenotype in macrophages and control the number and function of Tregs in adipose tissue through secretion of IL-2 (Lynch *et al*, 2015). Our experiments show that NKT cells contribute to the regulation of the intestinal Treg population through an IL-4-dependent mechanism. Although the role of IL-4 in the induction of Tregs remains controversial, several studies have shown the capacity of IL-4 to control Treg proliferation and/or survival (Skapenko *et al*, 2005; Pillai *et al*, 2009). Added to this, NKT cell-derived IL-4 has been shown to promote Treg function and to induce tolerance in combined bone marrow and organ transplant models and in murine models of graft-versus-host disease (Pillai *et al*, 2009; Hongo *et al*, 2012). Importantly, steady-state production of IL-4 by iNKT cells has been shown to regulate other immune cell populations including CD8$^+$ memory T cells and IgE production by B cells (Lee *et al*, 2015). Thus, we hypothesize that the accumulation of IL-4-producing NKT2 cells in the mLN may provide an IL-4-enriched milieu that may contribute to the control of the Treg population. Thus, while dysregulation of intestinal NKT cells has been associated with intestinal inflammation (Olszak *et al*, 2012), we propose that in steady-state conditions, iNKT cells contribute to the regulation of intestinal homeostasis by shaping the microbial and immune cell

populations and ultimately controlling the microbiota–immune system crosstalk.

# Materials and Methods

### Mice

WT C57BL/6 mice were purchased from Charles River. CD1d$^{-/-}$, CD1d$^{+/-}$, CD1d$^{+/+}$ and CD1d$^{fl/fl}$CD11c$^{Cre}$ were on a C57BL/6 background and bred under specific pathogen-free (SPF) conditions at King's College London. Nur77$^{GFP}$ mice were from Kristin A. Hogquist (University of Minnesota) and provided by Adrian Hayday (King's College London). All animal experiments were approved by the King's College London's Animal Welfare and Ethical Review Body and the United Kingdom Home Office (project licence 70/7907).

### Generation of CD1d$^{fl/fl}$ and CD1d$^{-/-}$ mice

CD1d$^{fl/fl}$ mice were generated by multi-site Red/ET recombination in an ES line of B6 origin as reported (Gaya *et al*, 2018). Briefly, exon 3 of the Cd1d1 gene was loxP-flanked and a PGK/neomycin cassette was introduced for *in vitro* selection of the recombined ES cells. This cassette, flanked by FRT sites, was removed by crossing the mice with a C57BL/6 mouse expressing Flp recombinase. CD1d$^{fl/fl}$ mice were crossed with CD11c-Cre mice to obtain CD1d$^{fl/fl}$CD11c$^{Cre}$ mice. Cre$^{-/-}$CD1d$^{fl/fl}$ (Cre$^-$) mice were used as control littermates. To generate CD1d$^{-/-}$ mice, male CD1d$^{fl/fl}$ mice were crossed with female PGK-Cre mice.

### *In vivo* administration of antibiotics, αGalCer, FITC–dextran and *Yersinia enterocolitica*

Mice were orally administered 200 µl of a cocktail of antibiotics (5 mg/ml ampicillin, 5 mg/ml gentamicin, 5 mg/ml neomycin, 5 mg/ml metronidazole and 2.5 mg/ml vancomycin (all from Sigma-Aldrich) in filtered drinking water) daily for 14 days. PP were collected for IgA sequencing.

αGalCer (BioVision) was administered by oral gavage (2 µg/mouse), and tissues were collected 8, 16 or 72 h after oral dosing.

Mice were immunized orally with $1 \times 10^9$ colony-forming units (CFU) of *Y. enterocolitica* subsp. *enterocolitica* (serovar 8 biovar 1; NCTC 10598; DSMZ, Germany), and 24 h later, mLN were collected, digested with collagenase (1.5 mg/ml) at 37°C for 20 min and filtered with a 45-µm cell strainer. After centrifugation at $3,400 \times g$ for 10 min, bacteria were resuspended in 100 µl sterile PBS and cultured in *Yersinia* selective agar (Oxoid). The number of CFU per organ was determined 16–20 h later.

### Microbiota studies

CD1d$^{-/-}$ and CD1d$^{fl/fl}$CD11c$^{Cre}$ mice were co-housed with WT littermates until weaning (3 weeks) and then isolated into individual autoclaved cages. Two consecutives litters from the same breeding pair were used. Samples were collected 3–4 weeks after weaning for intestinal microbiota analysis. The intestinal content of the distal 3 cm of the small intestine (ileum), excluding 1 cm proximal to the caecum, was obtained by manual extrusion. This ileum portion was then flushed with cold PBS and opened longitudinally. The inner surface (ileum wall) was scraped with a scalpel and washed with PBS. For oral αGalCer experiments, fresh stool pellets were collected before and 10 days after αGalCer administration. Samples were stored at −80°C until processing.

Bacterial DNA was obtained from stool, ileum content and ileum wall using QIAamp Fast DNA Stool Kit (Qiagen), following the manufacturer's instructions. The numbers of intestinal bacteria were determined by qPCR using SYBR Green Master kit (Bio-Rad) and the following primers: Eubacteria-F: 5′-ACTCCTACGGGAGGCAGCAGT-3′; and Eubacteria-R 5′-ATTACCGCGGCTGCTGGC-3′. For the standard curve, DNA from *Pseudomonas denitrificans* isolated from stool of a C57BL/6 mouse was amplified using the primers mentioned above and cloned into a TOPO vector (TOPO TA Cloning Kit for Sequencing; Invitrogen).

Sequencing of the V1–V3 region of the 16S rRNA gene was performed at MR DNA (www.mrdnalab.com, Shallowater, TX, USA) using the Illumina MiSeq sequencing platform. 16S rRNA sequences were processed with Mothur as we have previously described with some minor modifications (Schloss *et al*, 2009; Isaac *et al*, 2017). Briefly, sequences shorter than 500 bp that contained homopolymers longer than 8 bp with undetermined bases or with a quality average score < 30 were not included in the analysis. Sequences were aligned to the 16S rRNA gene using the SILVA reference alignment as a template. Potential chimeric sequences were removed using chimera.uchime program. To minimize the effect of pyrosequencing errors in overestimating microbial diversity (Huse *et al*, 2008), rare abundance sequences that differ in up to four nucleotides from a high abundant sequence were merged to the high abundant sequence using the pre.cluster option in Mothur. Since different numbers of sequences per sample could lead to a different diversity (i.e. more OTUs could be obtained in those samples with higher coverage), we rarefied all samples to the number of sequences obtained in the sample with the lowest number of sequences (i.e. 3,094 sequences), with the exception of the experiment shown in Fig 3E–G and Appendix Fig S7E in which 20,000 sequences per sample were utilized.

OTUs were identified using the average-neighbour algorithm. Sequences with distance-based similarity of 97% or higher were grouped into the same OTU. Each sequence was classified using the Bayesian classifier algorithm with a bootstrap cut-off of 60% (Wang *et al*, 2007). Classification was assigned to the genus level when possible; otherwise, the closest level of classification to the genus level was given, preceded by "unclassified". In order to compare the overall microbiota similarity in the relative abundance of the OTUs present in different intestinal samples (Fig 3A and E, and Appendix Fig S7D), the Yue–Clayton distance was calculated for every pair of samples. Subsequently, principal coordinates analysis was performed on the resulting matrix of distances between each pair of samples using Mothur.

Two-tailed Wilcoxon non-parametric test was applied to identify significant microbiota taxonomic differences among groups of mice. The FDR approach was applied to adjust for multiple hypothesis testing (Benjamini *et al*, 2001). Very low abundant taxa and OTUs (< 10 counts in the two groups of samples under comparison) were not included in the statistical analysis. Changes with a $P < 0.05$ and FDR < 0.25 were considered significant.

### IgA sequencing and faecal Ig detection

IgA sequencing was performed as previously described (Lindner *et al*, 2012). Briefly, PP were harvested and kept in RNAlater (Sigma-Aldrich) until processing. PP were homogenized using a TissueLyser II (20 Hz, 2 min; Qiagen), and RNA was extracted using RNeasy Mini Kit (Qiagen). cDNA synthesis was performed with iScript Select cDNA Synthesis Kit (Bio-Rad) using a mix of three primers to enrich in IgA transcripts (Lindner *et al*, 2012), and IgA heavy chain was amplified with Phusion High-Fidelity DNA Polymerase (New England Biolabs). IgA amplicons were cloned using TOPO TA Cloning Kit for Sequencing (Invitrogen), and DNA sequencing was performed by Eurofins MWG Operon (Germany). Sequences were subjected to quality control (Wu *et al*, 2010) before submitting to IMGT/HighV-QUEST (www.imgt.org/) for identification of *IGHV* genes.

For faecal IgA detection, fresh stool pellets were homogenized in PBS (100 μl/10 mg faeces) and spun at $400 \times g$ for 5 min to remove large particles. The supernatant was collected and spun 10 min at $8,000 \times g$ to separate bacteria-coating IgA (pellet) from soluble IgA (faecal supernatant) (Kawamoto *et al*, 2012). Free IgA levels were detected in the faecal supernatant after centrifugation at $21,000 \times g$ using Mouse IgA ELISA Quantitation Set (Bethyl). Serum IgA and IgG levels were quantified using IgA and IgG ELISA Quantitation Sets (Bethyl), respectively.

### Tissue preparation and flow cytometry

Peyer's patches, spleen, mLN, inguinal LN (iLN), liver and thymus were harvested and smashed through a 45-μm strainer. SI and colonic LP were processed as described (Saez de Guinoa *et al*, 2017). Briefly, small intestine (excluding Peyer's patches) and colon were flushed with cold PBS, opened longitudinally and incubated for 20 min at 37°C in HBSS with 1 mM EDTA and 5% FCS. The supernatant containing epithelial cells and intraepithelial lymphocytes was discarded, and the remaining tissue was incubated for 45 min at 37°C with collagenase (1.5 mg/ml) and DNase (100 μg/ml).

Flow cytometry analyses were performed in FACS buffer (PBS with 1% BSA and 1% FCS) using the antibodies listed in Appendix Table S1. PBS57-loaded CD1d tetramers (αGalCer:tet) were provided by the NIH Tetramer Core Facility. For transcription factor staining, cells were fixed and permeabilized with Foxp3/Transcription Factor Staining Buffer Set (eBioscience). Dead cells were detected with fixable viability staining (Biolegend). Flow cytometry data were collected on a FACSCanto II or LSR-II flow cytometer (both from BD Biosciences) and were analysed with FlowJo software (TreeStar).

For analyses of commensal antigen presentation, faecal material was collected from the ileum and caecum, homogenized in PBS and spun at $400 \times g$ for 5 min to remove large particles. The supernatant was collected and spun 10 min at $8,000 \times g$ to separate bacteria which were lysed by sonication on ice. Single-cell suspensions of mLN were prepared as above and incubated with the commensal extract for 2 h at 37°C, and the cells were analysed by flow cytometry as described above.

### Cell sorting and quantitative PCR

iNKT cells (TCRβ+αGalCer:tet+ cells), B cells (B220+), DCs (CD11chigh) or T cells (TCRβ+) were purified by cell sorting with a

FACSAria II (BD Biosciences). For analyses of whole tissue, mLNs were homogenized using a TissueLyser II (20 Hz, 2 min; Qiagen). RNA extraction was performed using RNeasy Mini Kit (Qiagen). cDNA was synthesized with iScript Select cDNA Synthesis Kit (Bio-Rad), and gene expression was determined with SYBR Green Master kit (Bio-Rad) and the primers included in Appendix Table S2. Reactions were run in a real-time PCR system (ABI7900HT; Applied Biosystems).

### Histological sections and RNAscope

Briefly, the last 3 cm of the terminal ileum was fixed in 10% neutral-buffered formalin (NBD) for 24 h, embedded in paraffin and cut into 4-μm sections. Sections were dewaxed and rehydrated and stained with haematoxylin and eosin (H&E). Goblet cells were identified with Alcian blue counterstained with far-red nuclear stain. Intestinal commensal bacteria were detected by *in situ* hybridization using the RNAscope 2.5 HD Assay-BROWN kit and the probe EB-16S-rRNA (Cat. #464461) (Advanced Cell Diagnostics). Images were acquired with a Nikon Eclipse 90i.

### Analyses of B-cell development, proliferation and class-switch recombination

Lymphocyte populations were analysed by flow cytometry in single-cell suspensions from spleen, PP, SI-LP, peritoneal cavity and bone marrow using combinations of antibodies as previously described (Chapman *et al*, 2013).

For proliferation and class-switch experiments, B cells were purified by negative selection from single-cell suspensions from spleen using magnetic separation B-cell isolation kit (Dynabeads; Invitrogen) according to the manufacturer's instructions. Purified cells (purity > 95%) were cultured for 4–5 days at $10^6$ cells/ml in RPMI supplemented with 10% FCS and LPS (10 μg/ml; Sigma), IL-4 (10 ng/ml; Biolegend), TGF-β (2 ng/ml; eBioscience), IL-5 (10 ng/ml; Biolegend) and/or anti-mouse CD40 antibody (5 μg/ml, HM40-3; Biolegend). For proliferation analyses, B cells were labelled with 5 μM CellTrace Violet (Molecular Probes) in PBS for 15 min at 37°C before culture. Flow cytometry analyses were performed in FACS buffer using the antibodies listed in Appendix Table S1.

**Expanded View** for this article is available online.

### Acknowledgements

This work was funded by the Medical Research Council (grant MR/L008157/1, to P.B.). J.S.d.G. and R.J. were supported by Marie Curie Intra-European Fellowships (PIEF-GA-2013-627391 and H2020-MSCA-IF-2015-703639). We acknowledge the NIH Tetramer Core Facility (contract HHSN272201300006C) for provision of CD1d tetramers, Facundo Batista (Ragon Institute) for conditional CD1d mice and Adrian Hayday (King's College London) for Nur77GFP mice.

### Author contributions

JSdG designed and performed research, analysed the data and wrote the manuscript; RJ designed and performed research and analysed the data; MG contributed to the generation of conditional CD1d-deficient mice; DK and DD-W assisted with IgA analyses; MJG and CU performed microbiota analyses and advised on microbiota-related experiments; PB supervised and designed research, analysed the data and wrote the manuscript.

## Conflict of interest

The authors declare that they have no conflict of interest.

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
