## [Review Process File · The EMBO Journal]

CD1d-MEDIATED LIPID PRESENTATION BY CD11c⁺ CELLS REGULATES INTESTINAL HOMEOSTASIS

Julia Sáez de Guinoa, Rebeca Jimeno, Mauro Gaya, David Kipling, María José Garzón, Deborah Dunn-Walters, Carles Ubeda & Patricia Barral

Review timeline:

Submission date:	8 June 2017
Editorial Decision:	20 July 2017
Revision received:	1 November 2017
Editorial Decision:	5 December 2017
Revision received:	15 December 2017
Accepted:	22 December 2017

Editor: Karin Dumstrei

Transaction Report:

1st Editorial Decision

20 July 2017

Thank you for submitting your manuscript to The EMBO Journal. Your study has now been seen by three referees and their comments are provided below.

As you can see from the comments below the referees find the topic interesting, but they are also bit divergent in their views about suitability for publication here. Referee #3 in addition to raising some technical and mechanistic concerns, is not convinced that the analysis provides enough novel insight. Referee #1 and 2 are more positive regarding this aspect. Having looked carefully at all the points raised I do find that the analysis adds important new insight and I am not as concerned as referee #3 is regarding that aspect. However, further work is needed to strengthen the findings. Should you be able to extend the analysis and revise the manuscript along the lines suggested by the referees then we would like to consider a revised manuscript.

When preparing your letter of response to the referees' comments, please bear in mind that this will form part of the Review Process File, and will therefore be available online to the community. For more details on our Transparent Editorial Process, please visit our website: http://emboj.embopress.org/about#Transparent_Process

Let me know if we need to discuss any points more specifically

Looking forward to seeing the revised version.

REFEREE REPORTS

Referee #1:

In this manuscript, Saez de Guinoa and colleagues investigate the role of DC in regulating intestinal homeostasis via CD1d-mediated lipid antigen presentation, as the molecular mechanisms regulating iNKT cell contribution to mucosal immunity remain poorly understood.

Using an as yet unpublished CD1d^{fl/fl} mouse model, the authors demonstrate that DC

mediate presentation of intestinal derived lipids to iNKT cells, regulating their homeostasis and activation. Furthermore, they show that DC-iNKT crosstalk controls bacteria and immune cell populations in the intestine and that mice with CD1d deficient DC show dysbiosis, altered IgA production and reduced frequency of intestinal regulatory T cells.

1) Although the conclusions are interesting, I feel that the system has not been fully characterized, particularly in relation to the cross-talk between iNKT cells with CD1d positive intestinal epithelial cells, which the Blumberg group has previously shown to mediate an important protective role in mucosal immunity. Indeed, while absence of MHC class II on conventional DC results in microbial dependent intestinal inflammation (Loschko et al), it is not clear what are the long-term consequences of the lack on CD1d on DC only and whether there is increased susceptibility to experimental colitis, (as observed when CD1d molecules are absent in the epithelium). Specifically, what are the long-term consequences of the observed dysbiosis? It would be important to challenge CD1d^{fl/fl} x CD11c CRE mice with oxazolone to assess the severity of oxazolone-induced colitis and compare it with wild type mice. In addition, the authors should isolate DC from mLN and SI-LP of the respective mice and show in ex-vivo assays whether they are able to stimulate the DN32 hybridoma or iNKT cell lines, as a consequence of their in vivo CD1d loading with bacterial antigens.

Specific comments:

2) There is a lack of detail in the strategy followed for FACS analysis, particularly with regard to gates and quadrants: in Figure 1, how was the Nur77 gate set up? Presumably on the non-GFP littermates, however it is not specified (and it is not clear in panel B, using WT mice: was the FMO control performed?). Is there different basal auto-fluorescence in T cells and iNKT from different organs to justify different position of the rectangle gate in panel A?

3) In Fig 1D, CD103+CD11b+ DC from mLN show the highest CD1d expression, but this is not the case in SI-LP (Suppl Fig 1E). It is not discussed whether this is a real finding, due to anatomical differences, and whether this is caused by minimal iNKT cell activation upon α GalCer injection (Suppl Fig 1A).

4) The authors show increased Nur77 expression in intestinal resident iNKT cells upon α GalCer injection, which is lost in DC-CD1d Cre⁺ mice. However, this is not done in the Nur77-GFP mice and it would be an important control to show. Were other activation markers investigated on the iNKT cells?

5) Also, it would be important to assess the ex-vivo basal presentation of lipids by intestinal DC, as an extension to Figure 1A. Indeed, the authors have not crossed Nur77GFP mice with the DC-CD1d Cre, so we do not know what the basal level of Nur77 expression would be in this system.

6) Interesting differences are shown between iNKT subsets in DC CD1d Cre⁺ mice as compared to Cre⁻ littermates. This was measured at one time in their life only. How is the overall frequency of iNKT cells changing from weaning to adulthood in these mice? We know that the microbial exposure in early life shapes iNKT cell frequency and function (Olszac et al, Nature), and this could be the ideal model to understand whether it is through intestinal DC.

7) Considering the variability between mice strains and that in Figure 2D the % of iNKT17 in WT is like in Cre⁺ in panel 2B, data relative to DC-CD1d Cre⁻ mice after α GalCer injection should be added to the figure.

8) Why there is no difference in the composition of the microbiota in the ileum wall, between CD1d^{+/-} and ^{-/-} (Fig 3A)? The authors should discuss these results in the context of previously published 16S rRNA sequence analysis of fecal samples of WT and CD1d KO mice performed by Nieuwenhuis et al.

9) Some of the taxa plotted in Fig 3C (33UC, 58 UC, 123 UC) are not in panel B. Are any taxa increased in CD1d deficient mice (as shown for WT post α GalCer)?

10) In Figure 6, in addition to frequency of Treg/iTreg absolute number should be plotted upon

α GalCer injection. Which cells produce TGF β in panel 6B? Is there any TGF β in CD1d deficient mice or in DC-CD1d-Cre⁺ mice upon α GalCer injection? The control experiment with DC-CD1d-Cre⁻ mice is missing from this figure. The effect of CD1d deficiency on Tregs remains an observation and the mechanism is speculative, and should be strengthened.

Minor comments:

Most of the figures plot data with n=3, one experiment. How many times were the experiments performed, and what was the overall tabulation of the data?

Some of the results investigating the microbiome confirm and extend previous finding by the Blumberg group on WT and CD1d KO mice, and this should be discussed more clearly.

The continuous change between CD1d KO and DC-CD1d mice is quite confusing for the reader, perhaps the figures could be more clearly labelled.

Referee #2:

EMBOJ-2017-97537

This manuscript by Sáez de Guinoa et al. describes how the CD1d1 molecule expressed by various cell populations influences commensal intestinal bacteria composition as well as mucosal immune homeostasis. The authors used the CD1d1fl/fl mice that they crossed with PGK-cre or CD11c-cre mice, as well as oral administrations of alpha-galactosylceramide.

From their studies, the authors conclude that CD1d-mediated presentation of (intestinal) lipids by dendritic cells (to iNKT cells) regulates intestinal homeostasis.

The manuscript is well-written and well-balanced, and contains an impressive amount of high quality work.

Major comments/questions:

1. CD1d1fl/fl x PGK-cre mice and CD1d1fl/fl x CD11c-cre mice appear to give somewhat different results throughout the manuscript. The authors should attempt to summarize these and highlight the key similarities and differences between the two in the discussion, and discuss it in the light of the published literature.

2. The finding from Figure 1A is interesting and the authors conclude/propose that "iNKT cells in those tissues are receiving TCR stimulation which could possibility originate from the recognition of commensal-derived lipids." The authors should perform experiments that directly address the contribution of the microbiota in the observed Nur77 expression.

3. In the same line of thought, throughout the manuscript, the authors imply that CD1d presents intestinal lipids (microbiota-derived) to iNKT cells, which explains their phenotypes/effects, but there is actually little evidence for that (see comment #2 above), and a lot of the assays involve alpha-galactosylceramide administration. Another potential explanation is that CD1d1 senses lipids (being microbiota-derived or administered alpha-galactosylceramide) and transduces signals within CD1d1-expressing cells, such as IECs or DCs. This retrograde signaling has been suggested by the Olszak and Blumberg 2014 study and others. Crossing the CD1d1fl/fl x CD11c-cre mice with Jalpha18^{-/-} mice to perform some key experiments would certainly strengthened the authors' conclusions. If these experiments are not doable within the scope of this work, I would suggest to clearly discuss this possibility and tone-down some of the conclusions.

4. The size of some of the samples is low. This is true for the microbiota analyses. The authors refer to "maternal transmission" a few times and cite the Ubeda and Pamer JEM paper. When 4-5 mice per group are use, one wonders how many different litters and/or breeding pairs these mice are from. Increasing sample size would strengthen the conclusions. The clustering in Figure 3A is not very convincing. Some of the IgA analyses contain even lower numbers (n=2 for Abx-recovered mice). It is not easy to draw meaningful conclusions.

5. The IgA and iTreg findings are interesting but the conclusion that "NKT cells regulate intestinal

immunity" appears a bit over-stated. The authors should attempt to measure intestinal immune responses (e.g. T cell responses to PMA/iono treatment) or use an intestinal inflammation model (DSS, oxazolone).

Minor comments/questions:

1. The findings on Figure 4 are interesting. Have the authors attempted to assess mucus production? Again, it would be interesting to discuss the differences observed between the two strains used.
2. SFB is often undetectable through sequencing strategies due to its overall low abundance, and RT-qPCR approaches using specific primers should be used.

Referee #3:

In their manuscript "CD1d-mediated lipid presentation by dendritic cells regulates intestinal homeostasis," Guinoa et al. investigate a role for cell specific CD1d expression on invariant NKT cell activation, control of microbiota and IgA and modulation of regulatory T cells. The authors present 3 major observations that CD1d depletion in CD11c positive cells alters invariant NKT activation in response to oral alpha galactosylceramide, the depletion of CD1d alters commensal microbial composition and IgA repertoire, and that CD1d modulates regulatory T cell homeostasis. Although the paper is well written, the role of CD1d on dendritic cells and the relationship between these 3 observations by the approaches the authors use is unclear and the authors do not rigorously investigate the mechanisms responsible for each of these observations in sufficient depth. In addition, a number of the observations lack sufficient novelty. In general, the authors also do not provide sufficient information about cell frequency versus absolute number or the effects of CD1d deficiency on local versus systemic iNKT populations and they do not take into account changes in overall T cell and B cell population dynamics in their experimental models.

Major Concerns

CD11c is not a specific marker for DCs but is also expressed by specific subsets of lymphocytes (including gamma delta T cells which are an important T cell population in the gut), B cells and macrophages which can express CD1d and hence cannot be used to specifically deplete CD1d exclusively in dendritic cells. Thus the authors' cannot definitely state that the defects observed in their CD11c-Cre-CD1d fl/fl animals are specifically attributed to the activity of DCs. This is pertinent throughout but especially so in the experiments that involve analyses of IgA repertoire.

The fact that alphagalactosylceramide can induce mucosal NKT cell activation as described is well known (see Courtney et al. 2009). This observation is thus not novel and is further limited by the use of CD11c-Cre mice as noted.

It has been previously shown that CD1d deficiency alters the microbial composition of commensal populations, that CD1d activation by alphagalactosyl ceramide affects microbial communities and that CD1d deficiency is associated with increased bacterial translocation (Nieuwenhuis et al. 2009), which limits the novelty of the work especially in the absence of mechanistic studies.

The observation that IgA repertoire is altered in CD1d deficient mice is interesting. But there is no description of the mechanisms or cell types involved in view of the limitations of the models used and the role played by the microbial dysbiosis observed.

The observation that CD1d lipid antigen presentation can influence regulatory T cell development or function is interesting. Given the dysbiosis observed in CD1d depleted animals the authors do not sufficiently explain whether this phenotype is due to specific depletion of CD1d in certain subsets or merely reflects differences in lymphocyte development due to changes in microbial composition.

1st Revision - authors' response

1 November 2017

Re: "**CD1d-MEDIATED LIPID PRESENTATION BY DENDRITIC CELLS REGULATES INTESTINAL HOMEOSTASIS**" by *J. Saez de Guinoa, R. Jimeno, M. Gaya, D. Kipling, MJ Garzon, D. Dunn-Walters, C. Ubeda & P. Barral.*

Reply to reviewers

We are very grateful for the positive and constructive comments provided by the reviewers and their thoughtful questions. We believe that the additional experiments performed have strengthened the manuscript and revealed important new insights into the intestinal NKT cell biology.

Referee #1 (R1)

In this manuscript, Saez de Guinoa and colleagues investigate the role of DC in regulating intestinal homeostasis via CD1d-mediated lipid antigen presentation, as the molecular mechanisms regulating iNKT cell contribution to mucosal immunity remain poorly understood. Using an as yet unpublished CD1d^{fl/fl} mouse model, the authors demonstrate that DC mediate presentation of intestinal derived lipids to iNKT cells, regulating their homeostasis and activation. Furthermore, they show that DC-iNKT crosstalk controls bacteria and immune cell populations in the intestine and that mice with CD1d deficient DC show dysbiosis, altered IgA production and reduced frequency of intestinal regulatory T cells.

1) Although the conclusions are interesting, I feel that the system has not been fully characterized, particularly in relation to the cross-talk between iNKT cells with CD1d positive intestinal epithelial cells, which the Blumberg group has previously shown to mediate an important protective role in mucosal immunity. Indeed, while absence of MHC class II on conventional DC results in microbial dependent intestinal inflammation (Loschko et al), it is not clear what are the long-term consequences of the lack on CD1d on DC only and whether there is increased susceptibility to experimental colitis, (as observed when CD1d molecules are absent in the epithelium). Specifically, what are the long-term consequences of the observed dysbiosis? It would be important to challenge CD1d^{fl/fl} x CD11c CRE mice with oxazolone to assess the severity of oxazolone-induced colitis and compare it with wild type mice. In addition, the authors should isolate DC from mLN and SI-LP of the respective mice and show in ex-vivo assays whether they are able to stimulate the DN32 hybridoma or iNKT cell lines, as a consequence of their in vivo CD1d loading with bacterial antigens.

We would like to thank R1 for suggesting these additional experiments as we agree they make a valuable contribution to our revised manuscript. We have performed experiments as suggested using cocultures with primary DCs and DN32 cells. Despite our best efforts we didn't detect DN32 activation (as measured by IL-2 ELISA) in these experiments. We believe that this is possibly due to technical limitations as CD1d on DCs will be probably loaded with a mixture of various lipids which may not be recognized with high enough affinity by the TCR of DN32 cells as to secrete IL-2 to be detected by ELISA. To overcome these technical difficulties, we have performed additional experiments *ex vivo* using primary cells from the mLN and demonstrated that CD1d expression on CD11c+ cells is required to induce iNKT cell activation in response to commensal-derived antigens (measured as Nur77 upregulation in primary iNKT cells). This new data is included in **Figure 11**.

We agree with R1 that challenging mice with oxazolone is an interesting experiment to perform. However, this manuscript focuses on immune responses and dysbiosis induced in homeostatic conditions and we think that colitis experiments are beyond the scope of the current manuscript to address experimentally in a timely fashion. In our animal facility, CD11c-Cre mice don't develop spontaneous colitis; although it is well known that such phenotypes are strongly dependent on the microbiota, so this may be different in other animal facilities.

2) There is a lack of detail in the strategy followed for FACS analysis, particularly with regard to gates and quadrants: in Figure 1, how was the Nur77 gate set up? Presumably on the non-GFP littermates, however it is not specified (and it is not clear in panel B, using WT mice: was the FMO control performed?). Is there different basal auto-fluorescence in T cells and iNKT from different organs to justify different position of the rectangle gate in panel A?

Yes, gates in Figures 1A and 1B have been set up with littermate GFP- control mice (1A) and with an FMO (1B). We have changed all our panels through the manuscript that now include the appropriate controls (**Figures 1A, 1B, 1H and Supplementary Figures 1A**). We thank R1 for this suggestion as we think that these new plots better depict our data.

3) In Fig 1D, CD103+CD11b+ DC from mLN show the highest CD1d expression, but this is not the case in SI-LP (Suppl Fig 1E). It is not discussed whether this is a real finding, due to anatomical differences, and whether this is caused by minimal iNKT cell activation upon aGalCer injection (Suppl Fig 1A).

Figures 1D and 1E show CD1d expression in the various intestinal DC/macrophage subsets in steady-state conditions. These different populations express different levels of CD1d in the tissues and this is a consistent phenotype in the tissues of WT B6 and CD1d^{fl/fl}CD11c^{Cre} Cre- mice. We have further quantified the CD1d expression levels in the various tissues and CD11c⁺ populations and included this data in the new **Supplementary Figure 1G**. As the reviewer points out these differences in CD1d expression could be due to the different environmental conditions in mLN and SI-LP as cells in the LP may be exposed to epithelial-derived cytokines and/or microbiota-derived metabolites that could affect CD1d expression levels. We thank the reviewer for pointing this out, and we have further discussed these results in the revised manuscript.

4) The authors show increased Nur77 expression in intestinal resident iNKT cells upon aGalCer injection, which is lost in DC-CD1d Cre+ mice. However, this is not done in the Nur77-GFP mice and it would be an important control to show. Were other activation markers investigated on the iNKT cells?

As suggested by R1 we have measured Nur77 upregulation in Nur77-GFP mice after aGalCer administration and included this new data in **Figure 1E**.

We have also investigated additional markers for iNKT cell activation after oral lipid administration and detected increased expression of CD69 by flow-cytometry and increased IL-4 and IFN- γ production by qPCR in mLN iNKT cells after aGalCer treatment. This new data has been included in the new **Figure 1D** and **Figure 6F**.

5) Also, it would be important to assess the ex-vivo basal presentation of lipids by intestinal DC, as an extension to Figure 1A. Indeed, the authors have not crossed Nur77GFP mice with the DC-CD1d Cre, so we do not know what the basal level of Nur77 expression would be in this system.

We agree with the reviewer that this is an important question and we have performed experiments that demonstrate ex-vivo the capacity of DCs to present commensal derived antigens (see answer 1). Moreover, to determine whether CD1d expression on DCs could control the basal level of Nur-77 expression we have stained intracellular Nur77 in iNKT cells from CD1d^{fl/fl}CD11c^{Cre} Cre+ and Cre- mice. Interestingly we detected a small but consistent decrease in Nur77 levels in iNKT cells from the mLNs (but not from the spleen) of Cre+ mice in comparison with Cre- littermates. This new data is included in **Supplementary Figure 2E**.

6) Interesting differences are shown between iNKT subsets in DC CD1d Cre+ mice as compared to Cre- littermates. This was measured at one time in their life only. How is the overall frequency of iNKT cells changing from weaning to adulthood in these mice? We know that the microbial exposure in early life shapes iNKT cell frequency and function (Olszac et al, Nature), and this could be the ideal model to understand whether it is through intestinal DC.

We thank R1 for this suggestion as we agree that this is an interesting observation. We have performed additional experiments analyzing iNKT cells in CD1d^{fl/fl}CD11c^{Cre} mice at the time of weaning (3 weeks). Importantly, we observed a decrease in NKT17 cells in the mLN of young mice as we have previously detected for adult mice. This new data has been included in new **Supplementary Figure 4A** of the revised manuscript.

7) Considering the variability between mice strains and that in Figure 2D the % of iNKT17 in WT is like in Cre+ in panel 2B, data relative to DC-CD1d Cre- mice after aGalCer injection should be added to the figure.

We agree with R1 that these are important experiments and we have performed the control experiments with CD1d^{fl/fl}CD11c^{Cre} Cre- mice. This new data confirmed our previous observations with WT B6 mice and has been included in **Figure 2G-H**.

8) Why there is no difference in the composition of the microbiota in the ileum wall, between CD1d^{+/-} and ^{-/-} (Fig 3A)? The authors should discuss these results in the context of previously published 16S rRNA sequence analysis of fecal samples of WT and CD1d KO mice performed by Nieuwenhuis *et al.*

Although we didn't detect differences in terms of PCoA for the ileum wall in CD1d^{+/+} and CD1d^{-/-} mice we did observe significant differences at the level of several bacterial families as shown in Figure 2D. We have discussed this data further and put it in the context with the findings of Nieuwenhuis *et al.*

9) Some of the taxa plotted in Fig 3C (33UC, 58 UC, 123 UC) are not in panel B. Are any taxa increased in CD1d deficient mice (as shown for WT post aGalCer)?

Figure 3B contains those OTUs that are highly abundant (>1%), while Figure 3C contains those OTUs that are significantly different between groups (some of them represent less than 1% of the microbiota and that is why they were not included in Figure 3B).

There were some OTUs whose relative abundance mean increased in CD1d-KO mice vs WT littermates but they did not reach significance (p=0.094) and that is why we did not include them in the figure.

10) In Figure 6, in addition to frequency of Treg/iTreg absolute number should be plotted upon aGalCer injection. Which cells produce TGFβ in panel 6B? Is there any TGFβ in CD1d deficient mice or in DC-CD1d-Cre⁺ mice upon aGalCer injection? The control experiment with DC-CD1d-Cre⁻ mice is missing from this figure. The effect of CD1d deficiency on Tregs remains an observation and the mechanism is speculative, and should be strengthened.

We thank the reviewer for his/her suggestions. We agree that the effect of CD1d expression on Treg induction is very interesting and we have performed additional experiments to strengthen this part of the manuscript. We have also performed the requested control with CD1d^{fl/fl}CD11c^{Cre} Cre⁻ mice (**Figure 6D-E**) and depicted the frequency and numbers of Tregs and iTregs in all our experiments (**Figure 6A-E, Supplementary Figure 10**).

To provide mechanistic insight into the processes regulating Treg induction we analyzed the pattern of cytokine secretion in the mLN after lipid antigen administration. In the original manuscript, we showed that oral lipids induced increased levels of TGF-β mRNA in the mLN, which we have now found to be absent in CD1d-KO and CD1d^{fl/fl}CD11c^{Cre} Cre⁺ mice (**Figure 6G**). Added to this, we now show that activated iNKT cells produce IL-4 in the mLN (**Figure 6F**) and, in line with this, IL-4-producing NKT2s accumulate in the mLN after oral lipid administration (Figure 2). Since both TGF-β and IL-4 can control the Treg population, we investigated whether they play a role in Treg induction after iNKT cell activation *in vivo*. Strikingly, our new data shows that while blocking TGF-β didn't affect Treg induction, administration of an αIL-4 blocking antibody reduced iTreg expansion in response to aGalCer administration (**Figure 6H-I**). Importantly, we also demonstrate that iNKT cells are the major producers of IL-4 in the mLN both in response to oral lipid administration (**Figure 6K**) and after stimulation with PMA/ionomycin (**Figure 6J**). All this new data (**Figure 6**) supports and further expands our previous observations and provides mechanistic understanding of the processes by which iNKT cells can participate in the control of the Treg population through production of IL-4.

Minor comments:

Most of the figures plot data with n=3, one experiment. How many times were the experiments performed, and what was the overall tabulation of the data?

We apologize for the confusion regarding the number of experimental data. All experiments have been performed at least 2 independent times. We have specified the number of experiments performed and plotted the individual mice used to tabulate the data in all the Figures throughout the manuscript.

Some of the results investigating the microbiome confirm and extend previous finding by the Blumberg group on WT and CD1d KO mice, and this should be discussed more clearly.

We thank R1 for this suggestion and have discussed our data in the context of previous reports

The continuous change between CD1d KO and DC-CD1d mice is quite confusing for the reader, perhaps the figures could be more clearly labelled.

We have changed the labeling as suggested throughout the manuscript and Figures. DC-CD1d mice are now called CD1d^{fl/fl}CD11c^{Cre} mice

Referee #2 (R2):

This manuscript by Sáez de Guinoa et al. describes how the CD1d1 molecule expressed by various cell populations influences commensal intestinal bacteria composition as well as mucosal immune homeostasis. The authors used the CD1d1fl/fl mice that they crossed with PGK-cre or CD11c-cre mice, as well as oral administrations of alpha-galactosylceramide. From their studies, the authors conclude that CD1d-mediated presentation of (intestinal) lipids by dendritic cells (to iNKT cells) regulates intestinal homeostasis. The manuscript is well-written and well-balanced, and contains an impressive amount of high quality work.

Major comments/questions:

1. CD1d1fl/fl x PGK-cre mice and CD1d1fl/fl x CD11c-cre mice appear to give somewhat different results throughout the manuscript. The authors should attempt to summarize these and highlight the key similarities and differences between the two in the discussion, and discuss it in the light of the published literature.

We thank R2 for his/her suggestion and have clearly stated in the discussion the differences in phenotype between CD1d-KO and CD1d^{fl/fl}CD11c^{Cre} mice. Moreover, as suggested by R1 and to prevent confusion, we have changed the labeling of the mouse strains and DC-CD1d mice are now labeled as CD1d^{fl/fl}CD11c^{Cre} mice throughout the manuscript and Figures.

2. The finding from Figure 1A is interesting and the authors conclude/propose that "iNKT cells in those tissues are receiving TCR stimulation which could possibility originate from the recognition of commensal-derived lipids." The authors should perform experiments that directly address the contribution of the microbiota in the observed Nur77 expression.

In an effort to test *in vivo* if Nur77 expression is regulated by commensals we have treated Nur77-GFP mice with antibiotics for 2 weeks and measured GFP expression in iNKT cells of treated and control mice (**Figure for referees not shown**). However, we didn't detect any changes in Nur77 expression in iNKT cells (or in any other GFP+ cells) within the intestinal compartment. We believe that this is possibly due to the fact that a considerable number of bacteria and bacterial products still persist after antibiotic treatment. Possibly the only approach to unequivocally address this question will be to generate germ-free Nur77-GFP mice, which at present is outside our experimental capabilities.

To overcome these technical difficulties, we have performed experiments *ex vivo* and demonstrated that commensals can induce Nur77 upregulation in primary iNKT cells and that this activation requires CD1d expression in CD11c+ cells. This new data is included in **Figure 11**.

3. In the same line of thought, throughout the manuscript, the authors imply that CD1d presents intestinal lipids (microbiota-derived) to iNKT cells, which explains their phenotypes/effects, but there is actually little evidence for that (see comment #2 above), and a lot of the assays involve alpha-galactosylceramide administration. Another potential explanation is that CD1d1 senses lipids (being microbiota-derived or administered alpha-galactosylceramide) and transduces signals within CD1d1-expressing cells, such as IECs or DCs. This retrograde signaling has been suggested by the Olszak and Blumberg 2014 study and others. Crossing the CD1d1fl/fl x CD11c-cre mice with Jalpha18-/- mice to perform some key experiments would certainly strengthened the authors'

conclusions. If these experiments are not doable within the scope of this work, I would suggest to clearly discuss this possibility and tone-down some of the conclusions.

We agree with R2 that we cannot discard that retrograde signaling may have some indirect effect in intestinal homeostasis and we have discussed this possibility in the revised manuscript (page 19). To further explore this question, we have phenotyped DCs from CD1d^{fl/fl}CD11c^{Cre} Cre⁺ and Cre⁻ mice by flow-cytometry and sorted cells from mLN and SI-LP for qPCR analyses of cytokine production. This data is included in the new **Supplementary Figure 2**. We didn't detect significant differences in any of the parameters tested for CD1d⁺ and CD1d⁻ DCs in steady-state conditions. Nonetheless we agree with R2 that we cannot fully discard that CD1d expression itself may affect DC functions and we have discussed this in the revised manuscript.

4. The size of some of the samples is low. This is true for the microbiota analyses. The authors refer to "maternal transmission" a few times and cite the Ubeda and Pamer JEM paper. When 4-5 mice per group are use, one wonders how many different litters and/or breeding pairs these mice are from. Increasing sample size would strengthen the conclusions. The clustering in Figure 3A is not very convincing. Some of the IgA analyses contain even lower numbers (n=2 for Abx-recovered mice). It is not easy to draw meaningful conclusions.

We apologize for the confusion regarding the number of experimental data. All experiments include a minimum of 3 mice per group and experiments have been performed at least 2 independent times. To avoid confusion, we have plotted all our data including the individual mice used for quantification in each of the experiments.

For microbiota analyses we have used 2-3 consecutive litters from the same breeding pair as previously reported (Ubeda et al, JEM). We would like to point out that the changes observed for microbiota and IgA are specific for the CD1d-KO and DC-CD1d strains. Accordingly, we have performed similar analyses for mice lacking CD1d on B cells (CD1d^{fl/fl}CD19^{Cre}) and intestinal epithelial cells (CD1d^{fl/fl}Villin^{Cre}) kept in the same animal facility and we haven't detected any significant differences in any bacterial community between Cre⁺ and Cre⁻ littermates from these strains (at the levels of order, family, genus or OTU in ileum content or wall; **Figure for referees not shown**). The same holds true for the IgA repertoire which is unaltered between littermates of the CD1d^{fl/fl}CD19^{Cre} or CD1d^{fl/fl}Villin^{Cre} strains (**Figure for referees not shown**).

5. The IgA and iTreg findings are interesting but the conclusion that "NKT cells regulate intestinal immunity" appears a bit over-stated. The authors should attempt to measure intestinal immune responses (e.g. T cell responses to PMA/iono treatment) or use an intestinal inflammation model (DSS, oxazolone).

As suggested by R2 we have performed PMA/ionomycin stimulation of cells from the mLN and SI-LP of CD1d^{fl/fl}CD11c^{Cre} Cre⁺ and Cre⁻ mice and measured secretion of cytokines by iNKT cells and conventional T cells. This new data is included in **Supplementary Figure 4B**. We thank R2 for suggesting these experiments, as they enabled us to gain mechanistic insights in the processes by which iNKT cell activation can control Treg induction. Interestingly, we have found that iNKT cells are the major producers of IL-4 within the mLN (both after stimulation with PMA/ionomycin and after aGalCer administration), and this IL-4 secretion is central to control Treg induction in response to oral lipid administration (**Figure 6**).

Minor comments/questions:

1. The findings on Figure 4 are interesting. Have the authors attempted to assess mucus production? Again, it would be interesting to discuss the differences observed between the two strains used.

We have measured mRNA levels for mucins and several anti-microbial peptides in purified IECs from WT/KO mice and included this new data in **Supplementary Figure 7C-D** and discussed it in the revised manuscript.

2. SFB is often undetectable through sequencing strategies due to its overall low abundance, and RT-qPCR approaches using specific primers should be used.

We thank the reviewer for this suggestion. We have performed qPCR for SFB in the intestinal

bacteria from WT and CD1d-KO mice. This data is included in **Supplementary Figure 6C**. The levels of SFB detected by this method were still very low (C_T values were always over 30), and no differences were found between CD1d^{+/-} and CD1d^{-/-} mice.

Referee #3 (R3)

In their manuscript "CD1d-mediated lipid presentation by dendritic cells regulates intestinal homeostasis," Guinoa et al. investigate a role for cell specific CD1d expression on invariant NKT cell activation, control of microbiota and IgA and modulation of regulatory T cells. The authors present 3 major observations that CD1d depletion in CD11c positive cells alters invariant NKT activation in response to oral alpha galactosylceramide, the depletion of CD1d alters commensal microbial composition and IgA repertoire, and that CD1d modulates regulatory T cell homeostasis. Although the paper is well written, the role of CD1d on dendritic cells and the relationship between these 3 observations by the approaches the authors use is unclear and the authors do not rigorously investigate the mechanisms responsible for each of these observations in sufficient depth. In addition, a number of the observations lack sufficient novelty. In general, the authors also do not provide sufficient information about cell frequency versus absolute number or the effects of CD1d deficiency on local versus systemic iNKT populations and they do not take into account changes in overall T cell and B cell population dynamics in their experimental models.

We thank the reviewer for his/her suggestions. We have ensured that all our figures include both frequency and absolute numbers of iNKT cells (**Figure 2**), Tregs (**Figure 6**), T and B cells (**Supplementary Figure 10**). We have also included data for the populations of splenic and thymic iNKT cells in our CD1d^{fl/fl}CD11c^{Cre} strain (**Supplementary Figure 3B**) as well as for T, B and Treg populations in steady-state and after oral lipid administration in mLN and spleen (**Supplementary Figure 10**).

Major Concerns

CD11c is not a specific marker for DCs but is also expressed by specific subsets of lymphocytes (including gamma delta T cells which are an important T cell population in the gut), B cells and macrophages which can express CD1d and hence cannot be used to specifically deplete CD1d exclusively in dendritic cells. Thus the authors' cannot definitely state that the defects observed in their CD11c-Cre-CD1d fl/fl animals are specifically attributed to the activity of DCs. This is pertinent throughout but especially so in the experiments that involve analyses of IgA repertoire.

We agree with the reviewer that CD11c-Cre doesn't only delete in DCs, and as such we pointed that out in our original manuscript where we showed CD1d deletion in DC and macrophages from DC-CD1d mice (Figure 1F-G). Importantly we didn't observe major changes on CD1d expression on B cells between Cre⁺ and Cre⁻ mice of our strain, nor in gdT cell that express very low levels of CD1d. We have included these controls in **Supplementary Figure 1E**.

The fact that alphagalactosylceramide can induce mucosal NKT cell activation as described is well known (see Courtney et al. 2009). This observation is thus not novel and is further limited by the use of CD11c-Cre mice as noted.

We agree with R3 in the appreciation that mucosal iNKT cells respond to oral aGalCer has been described and as such we have referred to several papers in our manuscript. However, the identity of the APCs involved in lipid presentation within the intestinal compartment and the effect on mucosal iNKT cell biology has never been investigated. Importantly, we not only show that CD11c⁺ cells are central to mediate iNKT cell responses to oral aGalCer, but they also control iNKT cell biology in steady-state conditions as demonstrated by the changes in the iNKT cell populations in the intestinal compartment of CD1d^{fl/fl}CD11c^{Cre} mice (Figure 2A and Supplementary Figure 3). Added to this, we also show that lipid presentation by CD11c⁺ cells drives the preferential expansion of NKT2 cells in the mLN (being this abrogated in Cre⁺ CD1d^{fl/fl}CD11c^{Cre} mice; Figure 2). These findings are particularly relevant in the context of Treg induction as we have now demonstrated that IL-4 produced by NKT cells in response to oral lipids is central in the control of the Treg population (Figure 6).

It has been previously shown that CD1d deficiency alters the microbial composition of commensal populations, that CD1d activation by alphagalactosyl ceramide affects microbial communities and

that CD1d deficiency is associated with increased bacterial translocation (Nieuwenhuis et al. 2009), which limits the novelty of the work especially in the absence of mechanistic studies.

We would like to reiterate that although changes in microbial communities in CD1dKO mice have been reported, the role of CD1d+ APCs in this phenotype has never been explored. Moreover, we have performed deep sequencing of the intestinal bacteria (with 20,000 reads per sample) which allowed us to perform in-depth analyses of the intestinal microbial communities in CD1d-KO and CD1d^{fl/fl}CD11c^{Cre} mice as well as in aGalCer-treated WT mice. Our results nicely complement and extend the data obtained by Nieuwenhuis *et al* who analyzed fecal samples by cloning and Sanger sequencing of intestinal bacteria in CD1d-KO mice (analyzing a maximum of 90 clones/mouse). Added to this, the changes in commensal communities induced by oral lipid administration have never been reported. Thus, we believe that our studies confirm the observations from Nieuwenhuis *et al* and further extend these results and provide insight into the cellular interactions mediating this phenotype.

The observation that IgA repertoire is altered in CD1d deficient mice is interesting. But there is no description of the mechanisms or cell types involved in view of the limitations of the models used and the role played by the microbial dysbiosis observed.

We have performed experiments (including B cell phenotyping in CD1d-sufficient and deficient mice, cholera toxin immunization, B cell proliferation and class-switch *in vitro*; Supplementary Figure 9) that demonstrate normal development and function for CD1d-deficient B cells. Moreover our *in vivo* experiments in which we induced changes of the microbiota (including antibiotic treatment and recovery and the use of mice from various animal facilities; Figure 5) demonstrate that the changes in IgA repertoire found in CD1d^{+/+} and CD1d^{-/-} mice arise as a result of changes in the intestinal microbiota from WT and KO mice and are not due to an intrinsic defect on B cell function.

Added to this, we have performed additional experiments with mice lacking CD1d expression on B cells (CD1d^{fl/fl}CD19^{Cre}) and we haven't detected any significant differences in any bacterial community between Cre+ and Cre- littermates from this strain (at the levels of order, family, genus or OTU in ileum content or wall; **Figure for referees not shown**). The same holds true for the IgA repertoire which is unaltered between littermates of the CD1d^{fl/fl}CD19^{Cre} strain (**Figure for referees not shown**).

All together our data demonstrates that changes in IgA repertoire are not due to an intrinsic defect in B cell development or activation but result from dysbiosis which arises from CD1d-deficiency.

The observation that CD1d lipid antigen presentation can influence regulatory T cell development or function is interesting. Given the dysbiosis observed in CD1d depleted animals the authors do not sufficiently explain whether this phenotype is due to specific depletion of CD1d in certain subsets or merely reflects differences in lymphocyte development due to changes in microbial composition.

We thank the reviewer for his/her suggestions. We agree that the effect of CD1d expression on Treg induction is very interesting and we have performed additional experiments to strengthen this part of the manuscript.

To provide mechanistic insight into the processes regulating Treg induction we analyzed the pattern of cytokine secretion in the mLN after lipid antigen administration. In the original manuscript, we showed that oral lipids induced increased levels of TGF- β mRNA in the mLN, which we have now found to be absent in CD1d-KO and CD1d^{fl/fl}CD11c^{Cre} Cre+ mice (**Figure 6G**). Added to this, we now show that activated iNKT cells produce IL-4 in the mLN (**Figure 6F**) and, in line with this, IL-4-producing NKT2s accumulate in the mLN after oral lipid administration (Figure 2). Since both TGF- β and IL-4 can control the Treg population, we investigated whether they play a role in Treg induction after iNKT cell activation *in vivo*. Strikingly, our new data shows that while blocking TGF- β didn't affect Treg induction, administration of an α IL-4 blocking antibody reduced iTreg expansion in response to aGalCer administration (**Figure 6H-I**). Importantly, we also demonstrate that iNKT cells are the major producers of IL-4 in the mLN both in response to oral lipid administration (**Figure 6K**) and after stimulation with PMA/ionomycin (**Figure 6J**). All this new data (**Figure 6**) supports and further expands our previous observations and provides mechanistic understanding of the processes by which iNKT cells can participate in the control of the Treg population through production of IL-4.

Thank you for submitting your revised manuscript to The EMBO Journal. Your study has now been seen by the original referees and their comments are provided below.

The referees appreciate that the analysis has been strengthened but also indicate that some further revisions are needed. Referee #1 offers a good suggestion to test further for the "quality" of iNKT cells in the Cd1d mice and I would like to ask you to carry out this experiment. Should be straightforward enough to do. Let me know if we need to discuss this further.

Referee #3 is still not persuaded by some of the findings. S/he raises the issue if CD11c is a good marker for DCs - perhaps a good solution for this issue is to refer to cells as "CD11c+ cells" instead of "DC". Let me know what you think about this. The other concerns can be address either in the point-by-point response or in the discussion section.

There are a number of minor points raised as well that will not involve too much additional work to sort out.

Given the input from the referees, I would like to invite you to submit a revised manuscript that addresses the raised points. I am happy to discuss everything further.

REFEREE REPORTS

Referee #1:

The authors have completed several new experiments, which have strengthened the manuscript. However, the evidence that the cross-talk between NKT cells and DC from mLN and SI-LP modulates gut microbiota is still indirect.

1) Although Figure S3 shows no differences in iNKT subtypes in Cre- and Cre+ mice, since the CD1d fl/fl mouse model is not an inducible KO mouse model, the authors need to assess whether the global lack of CD1d expression on CD11c positive cells may have a more general effect on the "quality" of NKT cells. Experiments demonstrating activation and cytokine profiling of NKT cells in the spleen upon intravenous (i.v.) injection of alphaGalCeramide (aGC) should be shown, as CD1d expression on B cells in the spleen/liver would contribute to NKT cell activation after i.v. injection of aGC.

Minor issues:

- 2) FiG 1H. The control with uninjected/vehicle injected mice is missing.
- 3) Fig. 1I: the FMO control is missing.
- 4). In Figs S1A, S2D and S2E there are no FMOs to control for the (low) Nur77 staining shown.
- 5). In Figs S1A, S2D and S2E there are no FMOs to control for the (low) Nur77 staining shown.
- 6). Fig 6K: GAPDH is not the correct control to normalize gene expression in DC, as GAPDH is metabolically regulated, and indeed the signal is lost upon DC activation. Furthermore, these data do not show that NKT are the main producers of IL4 after aGC injection, please address this issue in the manuscript.

Referee #2:

In this revised version of their manuscript, as well as in their rebuttal letter, the authors have addressed most of my comments and concerns raised from the original submission. The lack of effect of antibiotics treatment on Nur77 expression is somehow puzzling, but not entirely conclusive, as the authors pointed out. Short of re-deriving these mice in germ-free setting, it is difficult to conclude. But this would unreasonably delay the publication of this work.

This manuscript constitute a very thorough and mechanistic characterization (possibly the best to date) of the role of iNKT cells in the control of intestinal homeostasis.

Please carefully proofread the manuscript for typographical and grammatical errors.

In my opinion, the novelty and quality of this work meets the standard for publication in The EMBO Journal.

Referee #3:

Saez de Guinoa et al. Revised Manuscript, EMBO

This is a revised manuscript by Saez de Guinoa et al. entitled "CD1d-mediated lipid presentation by dendritic cells regulates intestinal homeostasis." In their manuscript, the authors utilize a newly developed CD1d fl/fl mouse model to investigate the activity presentation of lipids by dendritic cells (DC) and their effect on invariant NKT mediated responses during homeostatic and inflammatory conditions. They conclude the DC-iNKT interactions control immune response to bacterial derived antigens including alteration of IgA response and regulatory T cell development, however the mechanisms underlying the phenotypes they present and their interrelationships remain undeveloped. We believe the authors made some good faith but inadequate attempts to address the critique of their first submission, however the data presented is limited in novelty and unconvincing for publication.

Major Concerns.

CD11c: There is growing consensus that CD11c is an inappropriate marker of DCs despite the authors partial survey of select immune subsets to verify CD1d deletion using this mouse model. As a result it is possible (and likely) that the phenotypes the authors present in their model are not solely the influence of DC populations. This was brought up in the previous review and not adequately addressed.

Microbial Analysis and IgA: The current studies confirm and extend evidence for a role for CD1d expressing cells to influencing microbial composition. The new advance here is that CD11c+ cells expressing CD1d control microbiota as previously shown for total ko mice. Are the effects due to CD11c positive B cell populations given the leakiness of the Cre used or are they secondary to microbial effects caused by CD1d deletion in another cell type that is CD11c positive? In the absence of mechanism the results are descriptive and currently best described as potentially unrelated events. This could begin to be addressed in fecal transplant or co-housing experiments between WT and CD1d deficient animals. But it is best if the authors used a different set of Cre mice.

Treg: The authors demonstrate that regulatory T cell frequency in response to iNKT activation by CD11c+ cells depends on IL-4 activity. However, it is not clear what is the responsible CD11c+ cell involved and what is the role of altered microbiota observed? Is it primary (related to the CD1d positive cells) or secondary (due to the microbiota)?

2nd Revision - authors' response

15 December 2017

Reply to reviewers

Referee #1 (R1)

The authors have completed several new experiments, which have strengthened the manuscript. However, the evidence that the cross-talk between NKT cells and DC from mLN and SI-LP modulates gut microbiota is still indirect.

1) Although Figure S3 shows no differences in iNKT subtypes in Cre- and Cre+ mice, since the CD1d fl/fl mouse model is not an inducible KO mouse model, the authors need to assess whether the global lack of CD1d expression on CD11c positive cells may have a more general effect on the "quality" of NKT cells. Experiments demonstrating activation and cytokine profiling of NKT cells in the spleen upon intravenous (i.v.) injection of alphaGalCeramide (aGC) should be shown, as CD1d

expression on B cells in the spleen/liver would contribute to NKT cell activation after i.v. injection of aGC.

We have performed the additional experiments suggested by R1 and measured iNKT cell activation in spleen and liver from CD1dfl/flCD11cCre mice after i.v. administration of lipids. Interestingly, we found that CD1d expression on CD11c+ cells is important to mediate activation and cytokine secretion by splenic iNKT cells, as both Nur77 expression and cytokine secretion were strongly reduced in iNKT cells from Cre+ CD1dfl/flCD11cCre mice. However, CD1d expression on CD11c+ cells is dispensable for activation of liver iNKT cells, suggesting that different APCs may mediate lipid presentation in various tissues and/or in response to different routes of antigen administration. This data is included in the new Supplementary Figure 3A.

Minor issues:

2) *Fig 1H. The control with uninjected/vehicle injected mice is missing.*

We have included the control in Figure 1H.

3) *Fig.1I: the FMO control is missing.*

FMO has been included in Figure 1I

4). *In Figs S1A, S2D and S2E there are no FMOs to control for the (low) Nur77 staining shown.*

5). *In Figs S1A, S2D and S2E there are no FMOs to control for the (low) Nur77 staining shown.*

FMOs have been included in the Figures

6). *Fig 6K: GAPDH is not the correct control to normalize gene expression in DC, as GAPDH is metabolically regulated, and indeed the signal is lost upon DC activation. Furthermore, these data do not show that NKT are the main producers of IL4 after aGC injection, please address this issue in the manuscript.*

We thank the reviewer for pointing this out. We have now performed qPCR experiments using *HPRT-1* as reference gene and included this data in the revised Figure 6K. This new experiment showed similar results as the ones that we obtained previously (we didn't detect significant differences in IL-4 production after oral lipid administration in any of the populations tested). Nonetheless, we agree with R1 that we cannot discard that other cells may also contribute to IL-4 secretion in the mLN and we have stated this in the revised manuscript.

Referee #2 (R2):

In this revised version of their manuscript, as well as in their rebuttal letter, the authors have addressed most of my comments and concerns raised from the original submission.

The lack of effect of antibiotics treatment on Nur77 expression is somehow puzzling, but not entirely conclusive, as the authors pointed out. Short of re-deriving these mice in germ-free setting, it is difficult to conclude. But this would unreasonably delay the publication of this work.

This manuscript constitute a very thorough and mechanistic characterization (possibly the best to date) of the role of iNKT cells in the control of intestinal homeostasis.

Please carefully proofread the manuscript for typographical and grammatical errors.

In my opinion, the novelty and quality of this work meets the standard for publication in The EMBO Journal.

We thank the reviewer for his/her comments and his/her appreciation of our manuscript

Referee #3 (R3)

This is a revised manuscript by Saez de Guinoa et al. entitled "CD1d-mediated lipid presentation by dendritic cells regulates intestinal homeostasis." In their manuscript, the authors utilize a newly developed CD1d fl/fl mouse model to investigate the activity presentation of lipids by dendritic cells (DC) and their effect on invariant NKT mediated responses during homeostatic and inflammatory conditions. They conclude the DC-iNKT interactions control immune response to bacterial derived antigens including alteration of IgA response and regulatory T cell development, however the mechanisms underlying the phenotypes they present and their interrelationships remain

undeveloped. We believe the authors made some good faith but inadequate attempts to address the critique of their first submission, however the data presented is limited in novelty and unconvincing for publication.

Major Concerns

CD11c: There is growing consensus that CD11c is an inappropriate marker of DCs despite the authors partial survey of select immune subsets to verify CD1d deletion using this mouse model. As a result it is possible (and likely) that the phenotypes the authors present in their model are not solely the influence of DC populations. This was brought up in the previous review and not adequately addressed.

We agree with the reviewer that CD11c-Cre doesn't only delete in DCs and, as we show in our manuscript, CD1d is also deleted in macrophages from Cre+ CD1dfl/flCD11cCre mice (Figure 1F-G). To avoid confusion, we have changed the nomenclature throughout the manuscript and we now refer to CD11c+ cells instead of DCs to account for the heterogeneity of the CD11c+ population.

Microbial Analysis and IgA: The current studies confirm and extend evidence for a role for CD1d expressing cells to influencing microbial composition. The new advance here is that CD11c+ cells expressing CD1d control microbiota as previously shown for total ko mice. Are the effects due to CD11c positive B cell populations given the leakiness of the Cre used or are they secondary to microbial effects caused by CD1d deletion in another cell type that is CD11c positive? In the absence of mechanism the results are descriptive and currently best described as potentially unrelated events. This could begin to be addressed in fecal transplant or co-housing experiments between WT and CD1d deficient animals. But it is best if the authors used a different set of Cre mice.

As shown in Supplementary Figure 1E we haven't found major changes in CD1d expression on B cells in our CD1dfl/flCD11cCre strain. Moreover, we have performed additional experiments with mice lacking CD1d expression on B cells (CD1dfl/flCD19Cre) and we haven't detected any significant differences in bacterial communities or IgA repertoire between Cre+ and Cre- mice of this strain. Consequently, even if there is some minor deletion of CD1d on B cells in CD1dfl/flCD11cCre mice, this wouldn't account for the phenotypes observed in this strain.

Treg: The authors demonstrate that regulatory T cell frequency in response to iNKT activation by CD11c+ cells depends on IL-4 activity. However, it is not clear what is the responsible CD11c+ cell involved and what is the role of altered microbiota observed? Is it primary (related to the CD1d positive cells) or secondary (due to the microbiota)?

As suggested by R3, several CD11c+ populations could be involved in the activation of mLN iNKT cells and subsequent iTreg induction. Importantly, CD103+CD11b+ DCs have been described as a main migratory population that arrives to the mLN from the LP. We have shown that these cells express the highest levels of CD1d (Figure 1G) and could potentially transport and present lipid antigens to mLN iNKT cells. Nonetheless all of the CD11c+ cells in mLN and lamina propria express CD1d and they could be involved in lipid presentation in the various anatomical locations and/or at different stages of the immune responses. We have discussed this in the revised manuscript.

We agree with R3 that changes in microbiota could potentially influence the Treg population in our experiments. However, our data shows that Treg induction in response to NKT cell activation is dependent on CD1d-expressing CD11c+ cells (is absent on CD1dfl/flCD11cCre Cre+ mice) and on IL-4 (Treg induction is blocked with an aIL-4 antibody). Thus, it is the CD1d-dependent activation of NKT cells in the mLN that will ultimately control Treg induction.

Corresponding Author Name: Patricia Barral

Manuscript Number: EMBOJ-2017-97537R